# Molecular Correlates of Diapause in *Aphidoletes aphidimyza*

**DOI:** 10.3390/insects15050299

**Published:** 2024-04-23

**Authors:** Xiaoyan Dai, Yu Wang, Yan Liu, Ruijuan Wang, Long Su, Zhenjuan Yin, Shan Zhao, Hao Chen, Li Zheng, Xiaolin Dong, Yifan Zhai

**Affiliations:** 1Institute of Plant Protection, Shandong Academy of Agricultural Sciences, Jinan 250100, China; 15169087554@163.com (X.D.); abeamsunshine@163.com (Y.W.); liuyan8882@126.com (Y.L.); wangruijuan1020@126.com (R.W.); rizhaosulong@163.com (L.S.); zhaoshan0328@163.com (S.Z.); cha.active@163.com (H.C.); zhengli64@126.com (L.Z.); 2Key Laboratory of Natural Enemies Insects, Ministry of Agriculture and Rural Affairs, Jinan 250100, China; 3MARA-CABI Joint Laboratory for Bio-Safety Shandong Sub-Center, Jinan 250100, China; 4College of Agriculture, Yangtze University, Jingzhou 434023, China; dongxl@yangtzeu.edu.cn; 5College of Agriculture, Guizhou University, Guiyang 550025, China; yinzhenjuan1220@126.com

**Keywords:** *Aphidoletes aphidimyza*, diapause induction, sensitive stage, transcriptome, carbohydrate metabolism

## Abstract

**Simple Summary:**

*Aphidoletes aphidimyza* (Rondani) is an important predatory insect, primarily preying on aphids. The effective control of diapause in *A. aphidimyza* is a crucial practical issue in the field of biological pest control and large-scale artificial insect breeding. Previous research has shown that, in the European strain of *A. aphidimyza*, diapause occurs during the larval stage in the cocoon, it has a day-long response, and it enters diapause when the day length decreases. In this study, we evaluated the effects of photoperiod and temperature on the incidence of diapause in *A. aphidimyza*. We found that the diapause occurs in the cocooning larvae stage, and the highest diapause rate was recorded at under 15 °C and 10L:14D photoperiod conditions. Transcriptome sequencing analysis showed that differentially expressed genes were mainly enriched in the glucose metabolism pathway.

**Abstract:**

The aphidophagous gall midge, *Aphidoletes aphidimyza* (Rondani) (Diptera: Cecidomyiidae), a dominant natural enemy of aphids, has been used as a biological control agent in many countries to control aphids in greenhouses. To identify key factors that induce diapause in *A. aphidimyza*, we evaluated the effects of photoperiod and temperature on the incidence of diapause in *A. aphidimyza* under laboratory conditions. The results showed that temperature and photoperiod had significant impacts on development and diapause in *A. aphidimyza.* Low temperatures and a short photoperiod inhibited development, while high temperatures and a long photoperiod promoted development. Temperatures above 20 °C and a photoperiod greater than 14 h prevented diapause in *A. aphidimyza*. However, the highest diapause rate was recorded at under 15 °C and 10L:14D photoperiod conditions. At 15 °C, the first to third larvae were sensitive to a short photoperiod at any stage, and a short photoperiod had a cumulative effect on diapause induction. The longer the larvae received short light exposure, the higher the diapause rate appeared to be. Transcriptome sequencing analysis at different stages of diapause showed that differentially expressed genes were mainly enriched in the glucose metabolism pathway. Physiological and biochemical analyses showed that diapausing *A. aphidimyza* reduced water content; accumulated glycogen, trehalose, sorbitol, and triglycerides; and gradually reduced trehalose and triglyceride contents in the body with the extension of diapause time. Glycogen may be used as a source of energy, but sorbitol is usually used as a cryoprotectant. This study provided results on aspects of diapause in *A. aphidimyza*, providing data and theoretical support for promoting its commercial breeding and in-depth research on the molecular mechanisms underlying diapause regulation.

## 1. Introduction

*Aphidoletes aphidimyza* is a dominant predatory insect named after its larvae that causes plants to form galls. It is a monophage and one of the top ten most widely used arthropods in the biological control of aphids [1]. It has a wide distribution, a strong flight ability, the ability to quickly search for aphids, a strong predation ability, and the ability to predate multiple species of aphids, and it can easily breed [2]. *A. aphidimyza* is widely distributed throughout the Northern Hemisphere, including in Canada, 60° N of Russia, and 68 °N of Finland, as well as in Argentina, Chile, Brazil, and New Zealand [3]. In China, it is mainly distributed in Hubei and other regions [2]. *A. aphidimyza* preys on aphids on over 80 different crops [4]. Adult females effectively locate aphids over a relatively long range (up to 45 m) and lay eggs near or within the aphid population. The newly hatched larvae feed mainly by piercing one leg or other joints of the aphid’s body to inject toxic saliva to numb it or by piercing and sucking on its abdomen to draw fluids from the aphid’s body. The commercial production of *A. aphidimyza* began in the Netherlands in 1978. China introduced it from Canada in 1986. Now, they are mainly used for the prevention and control of vegetable aphids in facilities and protected areas, and this has achieved good control effects [5,6].

Development duration, temperature, and photoperiod have always been important parts of insect ecology. Different temperatures and photoperiods have different effects during the growth and development period of *A. aphidimyza*. Females live for about 4–9 days, and males live for about 3–5 days bred at 25 °C; adults can lay eggs the next day after mating; larvae hatch from the front of the egg and feed on aphids after 2–3 days; cocooned larvae prepare for pupation after 4–6 days; and the pupal stage is maintained for 8–12 days. Lin Qingcai’s research showed that, with an increase in temperature, the development rate of the eggs and larvae of *A. aphidimyza* accelerates [7]. Gong Yajun (2007) found that the optimal development temperature for *A. aphidimyza* is 19–23 °C, and development is significantly inhibited at 31 °C [8]. Therefore, the study of the effect of temperature and photoperiod on the developmental period of insects is of great significance in understanding the habits of insects and the release of natural enemies in the field.

The effective control of diapause in *A. aphidimyza* is a crucial practical issue in the field of biological pest control and large-scale artificial insect breeding. Environmental factors such as temperature and photoperiod, as well as genetic factors, influence insect diapause [9]. In the European strain of *A. aphidimyza*, diapause occurs during the larval stage in the cocoon [10]. This strain shows a day-long response and enters diapause when day length decreases. Havelka (1980) [11] reported that, for a European strain of *A. aphidimyza*, the most sensitive period for entering diapause is the non-cocooned third-instar larva and the cocooned larval stage. Masafumi et al. (2012) [12] reported the Japanese strain of *A. aphidimyza* is sensitive to diapause entry from the first to third non-cocooned instar larval stages. The critical day length for diapause induction was determined to be 12.7 h to 20 °C. The interactions between the effects of temperature and photoperiod have never been studied in *A. aphidimyza* in China. Understanding and managing diapause induction in *A. aphidimyza* is essential for optimizing its potential as a biological control agent for aphids.

Changes in the external environment result in dynamic changes in the levels of metabolic, physiological, and biochemical substances in insects [13]. Most insects mainly accumulate sugars, lipids, and alcohols to maintain normal life activities. Insect diapause is always accompanied by a series of metabolic changes, among which, the most dominant is sugar metabolism, which mainly provides energy for their life activities, including catabolism and anabolism. The main metabolic pathways include glycolysis, gluconeogenesis, the citric acid cycle, and the pentose phosphate pathway [14]. Yocum et al. (2011) [13] reported that the changes in physiological and biochemical substance levels within the bodies of insects can be utilized to distinguish between diapause and the termination of diapause.

The development of sequencing technology to capture de novo transcriptions has promoted in-depth research on the molecular mechanism underlying insect diapause. This not only enriches the species’ genetic database but also promotes research on insect species without reference genomes [15]. Insect diapause is a response to changes in environmental signals, leading to alterations in internal signaling pathways and the expression of specific genes [15]. Therefore, research on the transcriptome of insect diapause is essential to understanding the underlying mechanisms, including the biomolecules involved.

Currently, there are few reports on diapause in *A. aphidimyza*. Additionally, there is currently no reference genome for *A. aphidimyza*, which hinders molecular studies on regulatory mechanisms underlying its diapause. Therefore, we conducted this study to understand the molecular regulatory mechanisms underlying diapause induction in *A. aphidimyza*. The results provide relevant information on diapause in *A. aphidimyza*, which can be applied to extending the shelf life of goods during production, increasing their population when the cost of greenhouse breeding is high and the breeding speed is slow in winter, as well as improving their efficiency in controlling aphids in the field.

## 2. Materials and Methods

### 2.1. Insect Rearing

The tested insects, *A. aphidimyza* and *A. Craccivora*, were collected from pea crop fields in Jinan, Shandong Province, China (N 36°41′, E 116°54′), and were reared in insect-rearing cages in a glass greenhouse (26 ± 1 °C, 70 ± 5% RH, 14L:10D) at the Institute of Plant Protection, Shandong Academy of Agricultural Sciences (Shandong, China, N 36°41′, E 116°54′). Broad bean seedlings were used as host plants to breed *A. craccivora*, which was then used to breed *A. aphidimyza*. All insect-rearing procedures were performed at 25 °C, a relative humidity of 65~75%, and a photoperiod of 14 h light and 10 h dark (14L:10D).

### 2.2. Effects of Temperature and Photoperiod on Diapause Induction of A. aphidimyza

To evaluate the effects of temperature and photoperiod, experiments were conducted in an artificial climate chamber under 5 temperatures, 15, 18, 20, 25, and 30 °C, and 5 light cycles: 8 h light and 16 h dark (8L:16D); 10 h light and 14 h dark (10L:14D); 12 h light and 12 h dark (12L:12D); 14 h light and 10 h dark (14L:10D); and 16 h light and 8 h dark (16L:8D). The set-up was covered with 6 layers of black cloth to regulate the length of the photoperiod and to keep the relative humidity at 70% ± 5%. Emerged 24 h old adults of *A. aphidimyza* were placed in insect traps and placed in broad bean seedlings with aphids for laying eggs. After 24 h, the fresh eggs were transferred to a covered plastic cup with a soft brush. The cup wall was covered with mesh for ventilation, and wet filter paper was placed in it to provide moisture. The hatched larvae were then reared on 3rd instar aphid nymphs. The development of *A. aphidimyza* was observed daily to record their developmental duration. The experiments were conducted with 3 replicates, with 40 insects per treatment.

The diapausing status period of cocooning larvae of *A. aphidimyza* is divided into three categories: non-diapause, incomplete diapause, and complete diapause (Table 1) [16].

### 2.3. Insect Stage Sensitive to Photoperiod Induction of Diapause in A. aphidimyza

To determine the photoperiod-sensitive stage of *A. aphidimyza* that was most sensitive to photoperiods, experiments were conducted with eggs and cocooned larvae at 15 °C under two photoperiods, 10L:14D (S), and 16L:8D (L), in an artificial climate chamber (Table 2). Each treatment consisted of 50 test insects, with 3 replicates.

### 2.4. Transcriptomics Analysis and Quantitative Real-Time PCR (qRT-PCR) Validation

Diapausing and non-diapausing *A. aphidimyza* were placed under 10L:14D and 16L:8D at 15 ± 1 °C, respectively. The test insects were selected at the cocooned larvae stage (ND: non-diapause; D15: diapause for 15 days; D25: diapause for 25 days; D35: diapause for 35 days) for transcriptome sequencing (Table 3). The transcriptome sequencing analysis referred to the study by Yifan [16]. To verify the reliability of the transcriptome results, real-time quantitative PCR analysis was performed on selected genes (Appendix A). RNA was treated with DNase I (New York, NY, USA) to remove contaminated genomic DNA. cDNA was then synthesized using the Takara reverse transcription kit, which was diluted to 50 ng/µL. The cDNA obtained was used as a template to run a qPCR on the Quantstudio 6 Flex instrument using a 2×SYBR Green qPCR Mix (With ROX) kit (SparkJade, Shandong Sparkjade Biotechnology Co., Ltd., Jinan, China). The program was as follows: pre-denaturation at 94 °C for 3 min; denaturation at 94 °C for 15 s; annealing at 60 °C for 15 s; extension at 72 °C for 20 s; and collection of fluorescence through 40 PCR cycles. The relative level = 2^−ΔΔCT^ and means that the expression of the target gene is calculated by comparing the target gene with the internal reference gene.

### 2.5. Physiological and Biochemical Substance Determination

To elucidate the physiological adaptation of *A. aphidimyza* to diapause (10L:14D) and non-diapause (16L:8D) at 15 °C, metabolites related to cold tolerance and energy were measured using commercial kits; the samples were tested using whole insects and normalized according to total protein, which included water content (TRIzol Reagent, Ambion; Reverse Transcription Kit RR047A, TakaRa, Baori Doctor physical Technology Co., Ltd., Beijing, China), total lipids (Methanol, Tianjin Fourth Reagent Factory, Tianjin, China; Phosphate-Buffered Saline (1×PBS), Tianjin Enzyme Biotechnology Co., Ltd., Tianjin, China), triglycerides (Triglyceride (TG) Assay Kit, Nanjing Jiancheng Bioengineering Institute, Nanjing, China), glycogen (Glucose (GO) Assay Kit, Sigma-Aldrich, Shanghai, China), trehalose (Trehalose Assay Kit, Beijing Solabio Technology Co., Ltd., Beijing, China), and sorbitol (Sorbitol Assay Kit, Beijing Solabio Technology Co., Ltd., Beijing, China). Protein concentration was measured using the BCA protein quantification/concentration determination kit (Tianjin Enzyme Biotechnology Co., Ltd.).

### 2.6. Data Analysis

All statistical analyses were conducted using IBM SPSS statistics 23.0, and using Tukey’s test of variance, the results were expressed as mean ± standard error. Charts were created using GraphPad Prism 8.0.1.

## 3. Results

### 3.1. Effects of Temperature and Photoperiod on the Developmental Periods of A. aphidimyza

The results showed that different temperatures and photoperiods affect the developmental duration of *A. aphidimyza* (Figure 1). There were significant differences in the developmental duration at all stages under different photoperiods, except for the egg stage at 25 °C (*p* < 0.05). The developmental duration of the different life stages generally decreased with the extension of the photoperiod. The shortest developmental time of each stage was recorded under 16L:8D, and the longest developmental time of each stage was recorded under 8L:16D. Long photoperiods promoted development and shortened the developmental time at each stage. Conversely, short photoperiods inhibited development at each stage and prolonged the developmental time. There were significant differences in the developmental duration of *A. aphidimyza* at different stages under different temperature conditions (*p* < 0.05). The developmental time at each stage gradually decreased with increases in temperature. The longest developmental time at each life stage was recorded under 15 °C: egg stage (approximately 4–5 days), larval stage (approximately 10–12 days), cocooned larval stage (approximately 10–12 days), and pupal stage (approximately 15–20 days). The shortest developmental time at each stage was recorded at 30 °C: egg stage (about 2 days), larval stage (about 3 days), cocooning larval stage (about 3–4 days), and pupal stage (about 6–7 days). High temperatures promoted growth and development, while low temperatures, on the other hand, delayed the growth and development of *A. aphidimyza.*

Temperature and photoperiod had an interactive effect on the developmental duration of *A. aphidimyza.* There were significant differences (*p* < 0.05) in the duration of the eggs, larvae, cocooning larvae, and pupae of *A. aphidimyza* under the different combinations of temperatures and photoperiods (Table 3). The results showed that low temperatures and short light exposure inhibited the growth and development of *A. aphidimyza,* while high temperatures and long light exposure promoted them.

### 3.2. Effects of Temperature and Photoperiod on Diapause in A. aphidimyza

The effects of different temperatures and photoperiods on the diapause induction of *A. aphidimyza* are shown in Figure 2. Diapause mainly occurred within a range of 15–20 °C. The diapause rate significantly decreased with increasing temperature under the same photoperiod, with the highest diapause rate of 88.70% recorded at 15 °C. When the temperature increased from 15 °C to 18 °C, the diapause rate decreased to about 20% under 8–12 h of photoperiod. At 20 °C, the diapause rate slightly increased under 10 h of lighting, but there was no significant difference compared with 18 °C. The diapause rate gradually decreased under 8, 12, and 14 h of lighting. *A. aphidimyza* did not enter diapause at 25 or 30 °C under any photoperiod conditions. The diapause rate showed a trend of first increasing and then decreasing with an increase in photoperiod under the same temperature conditions. Diapause mainly occurred with lighting of 8 h to 14 h. *A. aphidimyza* did not undergo diapause under 16 h lighting at any temperature. At 15 °C and 20 °C, with 10L:14D, the diapause rate of *A. aphidimyza* reached the highest diapause. At 18 °C and 14L:10D, the diapause rate of *A. aphidimyza* was highest, but there was no significant difference compared with the diapause rate under the 10L:14D condition. The highest diapause rate was recorded under a 10L:14D photoperiod at 15 °C, 18 °C, and 20 °C.

Bivariate analysis showed that temperature and photoperiod had an interactive effect on the induction of diapause in *A. aphidimyza* (F = 2.997; df = 14, 60; *p* < 0.05). The results showed that, at the same temperature, short light exposure induced diapause in *A. aphidimyza*, but this was reversed under a longer light period (16 h). Under the same photoperiod conditions, low temperatures induced diapause in *A. aphidimyza*, but this was reversed under high temperatures.

### 3.3. Photoperiod Sensitive Stage of A. aphidimyza to Diapause Induction

The diapause rates of *A. aphidimyza* under different developmental stages under long and short light treatments at 15 °C are shown in Table 4 (F = 57.117; df = 9, 20; *p* < 0.05). The results showed that treatments 1–5 gradually increased the diapause rate under a cumulative short photoperiod at each life stage; the diapause rates were 16.5%, 59.38%, 83.33%, 100%, and 100%, respectively. This indicates that cumulative short photoperiod treatments at the first–third instar larvae positively correlated with diapause rates. The longer the cumulation of the short photoperiod, the higher the diapause rate. In treatments 6–10, a longer cumulative photoperiod gradually decreased the diapause rate in *A. aphidimyza*, with recorded diapause rates of 83.25%, 17.84%, 16.78%, 13.81%, and 0%, respectively; there was a difference in the diapause rate at the egg stage under different photoperiods when comparing treatments 1 and 10, as well as comparing 5 and 6. Likewise, there was no significant difference in the diapause rate at the cocooning larval stage under different photoperiods comparing treatments 4 and 5, as well as 9 and 10. This indicates that the egg stage and cocooning stage were not sensitive to light conditions. Therefore, the first–third instar larvae are the most sensitive to diapause induction under a short photoperiod.

### 3.4. Transcriptomic Analysis of A. aphidimyza

Assembling the obtained data using Trinity generated a total of 100,104 Unigenes, with a total length of 77,811,910 nt, an average length of 777.31 nt, and an N50 length of 1448 nt. The splicing length is also provided, indicating high-quality raw data splicing (Appendix A).

The Pearson correlation coefficient was determined for the gene expression levels of 12 samples (Figure 3). The high similarity and close correlation between the biological replicates indicated that the overall quality of the transcriptome data was high and was, therefore, usable for subsequent analysis. Due to the lack of a reference genome for *A. aphidimyza*, a total of 37,610 unigenes (37.57% of the assembled unigenes) obtained from the transcriptome results were functionally annotated by comparing them with nine major databases. The Unigenes annotated in each database were COG (8435), Swissprot (15,901), KOG (25,554), eggNOG (25,184), Pfam (26,363), GO (28,663), KEGG (29,481), TrEMBL (29,705), and Nr (30,815) (Appendix A).

### 3.5. Differential Expression Gene Analysis

The ND, D15, D25, and D35 of the cocooning larvae of *A. aphidimyza* were compared to identify differentially expressed genes (Figure 4A). The results showed that 1279 genes were differentially expressed in D15 compared with ND, of which 403 genes were up-regulated, and 876 genes were down-regulated. In D25 compared with ND, 1836 genes showed differential expression, of which 647 were up-regulated and 1189 were down-regulated. In D35, compared with ND, 1944 genes showed differential expression, of which 805 were up-regulated, and 1139 were down-regulated. In D25, compared with D15, 190 genes showed differential expression, of which 43 genes were up-regulated, and 147 genes were down-regulated. In D35, compared with D15, 835 genes showed differential expression, of which 380 genes were up-regulated, and 455 genes were down-regulated. In D35, compared with D25, 246 genes showed differential expression, of which 144 genes were up-regulated, and 102 genes were down-regulated. The down-regulated differentially expressed genes were higher than the up-regulated differentially expressed genes in all other treatment groups, except for D25 vs. D35.

A total of 2370 differentially expressed genes were compared among the ND vs. D15, ND vs. D25, and ND vs. D35 groups (Figure 4B). Among them, 778 differentially expressed genes were co-expressed between groups; of these, 228 genes were up-regulated, and 550 genes were down-regulated. There were 213 differentially expressed genes specifically expressed in D15 compared with ND, of which 85 genes were up-regulated, and 128 genes were down-regulated. In D25, compared with the ND comparison group, there were 356 differentially expressed genes, of which 163 genes were up-regulated, and 193 genes were down-regulated. In D35, compared with ND, there were 610 differentially expressed genes, of which 353 genes were up-regulated, and 257 genes were down-regulated.

A total of 992 differentially expressed genes were compared among the D15 vs. D25, D15 vs. D35, and D25 vs. D35 groups (Figure 4C). Among them, four differentially expressed genes were co-expressed between groups, with two genes up-regulated and two genes down-regulated. There were 62 differentially expressed genes in D25 compared with D15, of which 12 genes were up-regulated, and 50 genes were down-regulated. There were 574 differentially expressed genes in D35 compared with D15, of which 270 genes were up-regulated, and 304 genes were down-regulated; 81 differentially expressed genes were specifically expressed in D35 compared with D25, with 49 genes up-regulated and 32 genes down-regulated.

### 3.6. GO Enrichment Analysis of Differentially Expressed Genes

GO functional annotation was performed on the differentially expressed genes obtained by pairwise comparison between the non-diapause group and the three different diapause groups at different times (Figure 5). The results showed that ND vs. D15d, ND vs. D25d, ND vs. D35d, D15d vs. D25d, D15d vs. D35d, D25d vs. D35d, and D25d vs. D35d each annotated 809; 1129; 1198; 113; 545; and 151 differentially expressed genes enriched into the three major categories of GO functions, mainly including biological processes, cell composition, and molecular functions. Pairwise comparison at each stage showed that the differentially expressed genes were mainly enriched in the cells, membranes, and organelles of the cell composition; cellular process, metabolic process, and single-organism process were more abundant in biological processes; binding, catalytic, and transporter activities were more abundant in molecular functions.

The significantly enriched entries (*p* < 0.05) were mainly related to metabolic and cellular processes in the six groups of comparisons, with the first 20 GO entries detailed in ribosome biogenesis, the integral component of membrane, and sequence-specific DNA binding (Appendix A).

### 3.7. KEGG Enrichment Analysis of Differentially Expressed Genes

KEGG enrichment analysis was performed on differentially expressed genes obtained by comparing non-diapause and diapause treatments at different times. According to the KEGG Orthology classification, they were enriched in 6 primary metabolic pathways; 28, 33, 35, 22, 35, 25 secondary metabolic pathways; and 144, 146, 158, 66, 174, and 87 tertiary metabolic pathways. The metabolic signaling pathway accounted for the largest proportion of genes in the enriched primary metabolic pathway (Figure 6A). The highest proportion of genes in the secondary metabolic pathway was carbohydrate metabolism (Figure 6B). The 13 tertiary pathways enriched in carbohydrate metabolism mainly included the citrate cycle (TCA cycle), fructose and mannose metabolism, galactose metabolism, glycolysis/gluconeogenesis, amino sugar and nucleoside sugar metabolism, inositol phosphonate metabolism, pentose and glucuronate interconversion, the pentose phosphonate pathway, pyruvate metabolism, and starch and sucrose metabolism. These involved 36 differentially expressed genes, of which 4 genes were up-regulated, and 32 were down-regulated.

The significantly enriched metabolic pathways (*p* < 0.01) were mainly related to lifespan regulation pathways, galactose metabolism, starch and sucrose metabolism, glucose metabolism, and degradation in all six comparisons. Detailed information on the top 20 significantly enriched KEGG signaling pathways is shown in the Appendix A.

### 3.8. Real-Time Quantitative PCR Analysis

Based on our analysis of differentially expressed genes, key genes in carbohydrate metabolism such as starch and sucrose metabolism, the pentose phosphate pathway, glycolysis/gluconeogenesis, and the citrate cycle metabolism pathway were selected for real-time quantitative PCR (qRT-PCR) validation (Figure 7). The DNA sequences have been provided in the Appendix A. The qPCR results showed consistent expression with that of the transcriptome, indicating the high accuracy of the transcriptome data. Further analysis of these genes revealed that HK, PFK, LDH, PEPCK, GP, CS, G6P, G6PI, TPP, and TREA showed lower expressions in diapause, while the other genes showed higher expressions. The expression levels of most genes gradually decreased with the prolongation of diapause.

### 3.9. Water and Fat Contents in A. aphidimyza

The water, total lipid, and triglyceride contents in *A. aphidimyza* at different diapause stages are shown in Figure 8. The results showed that there were significant differences in the free water content in the bodies of *A. aphidimyza* at different diapause stages (F = 17.249; df = 3, 8; *p* < 0.05). The water content of non-diapausing individuals was significantly higher than that of diapausing ones, by 11%. After entering diapause, the water content remained at a relatively stable level in *A. aphidimyza* (Figure 8A).

There was no significant difference in total lipid content between different diapause stages (F = 2.253; df = 3, 8; *p* = 0.159). However, the non-diapausing stage was higher compared with the diapausing stage. As the diapause time prolonged, the total lipid content gradually decreased, but there was no significant difference compared with the total lipid content in the non-diapause stage (Figure 8B).

There were significant differences in triglyceride content among different diapausing stages of *A. aphidimyza* (F = 12.078; df = 3, 32; *p* < 0.05). At the beginning of diapause, the triglyceride content was 160% higher than that of the non-diapause period. With the extension of diapause time, although the triglyceride content gradually decreased, it was still higher than that of the non-diapause period (Figure 8C).

### 3.10. Glycogen, Trehalose, and Sorbitol Contents in A. aphidimyza

There were significant differences in the trehalose content of different diapausing stages of *A. aphidimyza* (F = 37.242; df = 3, 38; *p* < 0.05). There was a difference in glycogen content between diapause days 15 and 35, which was 34% and 52% higher than on the non-diapause days. The trehalose content decreased by 62% on diapausing days 25 and 35 compared with non-diapause days. The trehalose content significantly decreased with prolonged diapause time. Although the relative trehalose content slightly increased on diapause day 35, it was not significantly different compared with diapause day 25 (Figure 9A). There were significant differences in the glycogen content between different diapause stages of aphid-eating gall midges (F = 9.618; df = 3, 23; *p* < 0.05). The relative glycogen content at non-diapause was lower than that at diapause. There was a significant difference in the glycogen content between diapause days 15 and 35, which was 34% and 52% higher than that at the non-diapause stage. There was no significant difference when compared with diapause day 25. During diapause, the relative glycogen content showed a trend of first decreasing and then increasing, reaching its lowest level on diapause day 25 and then significantly increasing on day 35 (Figure 9B).

There were significant differences in sorbitol content during different diapause stages in *A. aphidimyza* (F = 119.298; df = 3, 16; *p* < 0.05). The relative sorbitol content at non-diapause was lower than that at diapause, and although it was 42% lower than at diapause day 15, there was no significant difference. The change in sorbitol content during diapause gradually increased with the prolongation of time and showed significant differences. Compared with non-diapause, the sorbitol content increased by 90% and 290% on diapause days 25 and 35, respectively (Figure 9C).

## 4. Discussion

### 4.1. Effects of Temperature and Photoperiod on the Developmental Duration of A. aphidimyza

Temperature and photoperiod affect the growth and development of insects. For example, a study showed that high temperatures promote the growth and development of *Diarthronomyia chrysanthemi* Ahlberg [17]. Temperature has also been shown to have a significant impact on the development of *A. aphidimyza* larvae [11]. Further, the developmental duration of *Epicauta gorhami* Marseul has been shown to gradually decrease with increasing temperature [18]. The results of this study showed that temperature and photoperiod have significant effects on the development of *A. aphidimyza*. High temperature and long light promote its growth and development, and low temperature and short light inhibit its growth and development. The diapause temperature and photoperiod for *A. aphidimyza* were 15 °C and 10L:14D. These results are consistent with other studies on the effects of different temperatures on the developmental duration of *A. aphidimyza* [7,19].

### 4.2. Effects of Temperature and Photoperiod on Diapause Induction of A. aphidimyza

Insects generally diapause to avoid the dangers posed by adverse environmental conditions. Many factors, including photoperiod, temperature, natural enemies, food, and humidity, influence insect diapause. Temperature and photoperiod are key environmental factors that have been found to induce diapause in insects [20]. For instance, *Harmonia axyridis* has been reported to enter diapause at 20 °C and 10L:14D [21]. A temperature of 15 °C and 9L:15D will inhibit the oviposition of *Chrysoperla sinica* [22]. A short photoperiod treatment for *Locusta migratoria* adults and low-temperature treatment for eggs induce diapause [23]. Therefore, it is crucial to explore the effects of temperature and photoperiod on insect diapause. Wu (2013) [24] reported the effect of temperature and photoperiod on diapause in insects such as *Harmonia axyridis*, *Colaphellus bowringi*, *Agrypon flexorium*, and *Lysiphlebus testaceipes*. Our results showed that *A. aphidimyza* can enter diapause in the form of cocooned larvae, which are affected at 15 °C and under 10L:14D.

Our study showed that the diapause rate in *A. aphidimyza* is inversely proportional to temperature and photoperiod. It gradually increased with a decrease in temperature. For instance, at temperatures above 20 °C, the cocooned larvae of *A. aphidimyza* did not enter diapause under either long or short photoperiods. Cocooned larvae of *A. aphidimyza* did not enter diapause under a photoperiod of 16 h. At 15 °C and under 10L:14D conditions, the diapause rate exceeded 88%. Our results are different from other studies on diapause in *A. aphidimyza* [12,25], in that the diapause conditions were 15 °C and 10L:14D, while others have reported that the diapause conditions are 18 °C and 8L:16D, which may be related to geographical locations. The critical length of diapause in *A. aphidimyza* has been reported to vary at different latitudes and decreases with decreasing latitude [10]. Moreover, research on *Apolygus lucorum* found that the diapause rate of different populations gradually decreases with an increase in geographical latitude [26,27]. A two-factor analysis of variance of temperature and photoperiod showed that they synergistically affect diapause in *A. aphidimyza*.

This study successfully induced diapause in *A. aphidimyza*, elucidating the temperature and photoperiod requirements for diapause induction in this mosquito species. These findings provide technical support and theoretical guidance for the artificial control of diapause, extending the shelf life of products and enhancing the effectiveness of applications in pest control. This research is of significant importance for the commercial production and large-scale application of *A. aphidimyza*.

### 4.3. Life Stage of A. aphidimyza Sensitive to Diapause Induction by Photoperiod

Insects exhibit significant differences in their sensitivities to diapause induced by photoperiod, and this is limited to a certain developmental stage [28]. Insects that undergo embryonic diapause are mainly sensitive to the photoperiod during the maternal stage. For example, when raised under normal conditions, *A. lucorum* produced non-diapausing eggs, but the exposure of the first instar larvae to a short photoperiod resulted in the production of diapausing eggs [29]. Insects that undergo larval diapause are mainly sensitive to the photoperiod throughout the entire larval stage or at a certain larval stage. For example, *Laodelphgax striatellus* undergoes larval diapause, and its third instar nymph is the most sensitive to the photoperiod [30]. Insects that undergo diapause in the pupal stage are sensitive to the photoperiod during the larval stage. For example, *Helicoverpa armigera* and *Antherea pernyi* are sensitive to the photoperiod during the third–fifth larval stages [31,32]. Insects that diapause as adults are mainly sensitive to the photoperiod in their larval or early pupal stages, such as *Orius nagaii*, which is the most sensitive to the photoperiod in its third–fifth instar larval stages [33]. Our results indicated that *A. aphidimyza* is most sensitive to the photoperiod during the first–third larval stages, with the egg and cocooned larval stages less sensitive to the photoperiod. Moreover, cumulative exposure under a short photoperiod promoted the diapause rate. The diapause rate of the third instar larvae reached 100% when exposed to cumulative short photoperiods. The reverse was observed under cumulative exposure to long photoperiods. The cumulative exposure of the first instar to a long photoperiod decreased the diapause rate from 83.25% to 17.84%, which was further lowered for cocooned larvae (0%). Our results are consistent with the photoperiod regulation of diapause exhibited by insects such as *Grapholitha molesta*, *Diaphania pylori* Walker, *Chrysopa palens*, *Microplitis mediator*, and *Chouioa cunea* Yang [34,35,36,37,38].

### 4.4. Transcriptome Sequence and Analysis of Differentially Expressed Genes in the Transcriptome

The results of the comparison in the Nr database showed a closer homologous relationship between most genes of *A. aphidimyza* and *C. nasturtii* [39]. Annotation results from the COG, KOG, and eggNOG, databases were dominated by modifications; protein conversion and chaperones; signal transduction mechanisms; and general function prediction. However, the most annotated result from the eggNOG database was an unknown function, accounting for 21.52%. This may be due to the lack of annotation information in the database for some genes with low expression levels.

The GO enrichment analysis of differentially expressed genes revealed that many functions were enriched in biological processes, mainly related to developmental morphology and metabolic processes. Differentially expressed genes were significantly enriched in the lifespan regulation pathway, possibly due to reduced physiological activity during insect diapause, a slower growth rate, and prolonged developmental duration. KEGG Orthology classification also showed that differentially expressed genes were significantly enriched in carbohydrate metabolism, with the highest proportion being tertiary metabolic pathways including fructose and mannose metabolism; galactose metabolism; glycolysis/gluconeogenesis; amino and nucleotide sugar metabolism; pentose phosphate pathway; the citric acid cycle (TCA cycle); pyruvate metabolism; and starch and sucrose metabolism. This indicates that the carbohydrate metabolism signaling pathway in *A. aphidimyza* undergoes significant changes during diapause, which may be influenced by short photoperiods and low temperatures. *A. aphidimyza* may accumulate carbohydrates and synthesize cryoprotectants to maintain energy metabolism and enhance cold tolerance. These levels may gradually change during diapause. The expression of genes related to carbohydrate metabolism pathways such as glycolysis and gluconeogenesis has been reported in different stages of diapause in silkworms [40]. Also, the activities of four metabolic enzymes in *Sitodiplosis mosellana* significantly increased at diapause [41].

### 4.5. Analysis of Differentially Expressed Genes in Glucose Metabolism Pathways

qRT-PCR validation showed that the expressions of the genes of the two rate-limiting enzymes HK and PFK and the metabolic enzyme PGK in the glycolysis pathway were consistent with those in the transcriptome. Of these, HK and PFK showed lower expression in diapausing *A. aphidimyza* compared with non-diapausing insects. PGK showed higher expression but decreased with prolonged diapause time. PGK catalyzes the production of ATP from phosphoenolpyruvate to pyruvate. Therefore, we hypothesize that glycolytic metabolism was inhibited during diapause, reducing metabolic activity and energy consumption. *Helicoverpa armigera* experiences the inhibition of glycolysis during diapause [42], with lower expression of HK compared with non-diapause individuals. Additionally, in the diapause stage of the parasitic wasp *Exorista civilis*, the expression of HK and PK in the glycolysis/gluconeogenesis pathway is up-regulated [43]. The results indicated that the regulation of the enzymes of insects entering diapause is different.

The genes of the rate-limiting enzymes PEPCK, G6P, and pyruvate carboxylase (PC), which are involved in gluconeogenesis, were also identified through transcriptome sequencing and qRT-PCR validation. The results showed that the PEPCK and G6P genes had lower expressions in diapausing *A. aphidimyza*, especially PEPCK, which showed a continuous downward trend during diapause. Therefore, we hypothesize that gluconeogenesis metabolism is also inhibited during diapause. The gluconeogenesis and glycolysis pathways are interrelated and may have been mutually inhibited. The simultaneous inhibition of the gluconeogenesis and glycolytic metabolic pathways in this study may be due to different substrate concentrations involved in the metabolism of both pathways. The results were consistent with the expression trend of PEPCK in diapausing *E. civilis*, *N. vitripennis*, and *A. gifuensis* [43,44] but differed from the expression trends in diapausing individuals of *A. albopictus*, *Drosophila*, and *Bombyx mori* [45,46]. In these species, the expression of PEPCK during diapause is higher compared with non-diapause states, suggesting that they may store more energy through the gluconeogenesis pathway to meet the requirements of growth and development.

The citric acid cycle is the main pathway in organisms for energy [40]. Many intermediate metabolites such as oxaloacetic acid, citric acid α-Ketoglutaric, succinic acid, fumaric acid, and malic acid are produced to promote gluconeogenesis [47]. Transcriptome sequencing and qRT-PCR validation showed that these six genes were up-regulated at diapause in *A. aphidimyza*, indicating that they may have promoted the production of a large amount of oxaloacetate and high citrate cycle activity, which may be related to the large amount of energy required for insects to enter diapause. Insects require a large amount of oxygen when entering diapause. During the whole diapause process, the respiratory metabolic rate of diapause insects usually shows a U-shaped curve. For example, in the whole process of *Pieris melete* and *H. armigera* diapause, the respiratory metabolic rate is relatively high in the early diapausing stage, the lowest in the diapausing stage, and increased again after the removal of diapause [48,49]. With diapause progression, the requirement for oxygen decreases, and its expression gradually decreases, which, in turn, may cause a decrease in citric acid cycle activity. This is consistent with the decrease in citric acid cycle activity in the diapausing pupae of *H. armigera* [40].

TPS and TPP are involved in trehalose biosynthesis [50]. The hydrolysis of trehalose relies on trehalase [51], which is the only hydrolytic enzyme that hydrolyzes trehalose into two molecules of glucose, providing raw materials for glycolysis [52]. Transcriptome sequencing and qRT-PCR validation showed that the synthase TPS in trehalose metabolism had significant up-regulation during diapause in *A. aphidimyza* compared with non-diapause. TPP showed down-regulation on diapause day 15 and was subsequently up-regulated. The results showed that the accumulation of trehalose was promoted by the diapause of *A. aphidimyza*. Glycogen and trehalose can be converted into each other, and glucose-6-phosphate (G6P) is the key enzyme involved in this conversion [53]. Glycogen synthase (GYS) and glycogen phosphorylase (GP) play major roles in glycogen metabolism. GYS showed a significant upregulation trend at diapause in *A. aphidimyza*, but its expression gradually decreased with the prolongation of diapause time. GP was up-regulated at non-diapause, promoting glycogen hydrolysis to produce glucose. During the diapause period, diapause insects need to accumulate enough energy to cope with adverse environmental conditions in order to ensure their survival and reproduction [49,54]. Research has shown that *A. aphidimyza* accumulates a large amount of glycogen, which is an energy reserve substance that can provide the required energy. The results were consistent with the expression trend of PEPCK in diapausing of *E. civilis*, *N. vitripennis*, and *A. gifuensis*.

### 4.6. Changes in Physiological and Biochemical Substances

Our study found that the water content in the body of *A. aphidimyza* was significantly lower at diapause than at non-diapause, maintaining a stable level during diapause. Most insects entering diapause in the Northern Hemisphere generally adapt to low temperatures by reducing their supercooling and freezing points to improve their cold tolerance [55,56]. Water content can affect the supercooling point and freezing point inside the insect body [57]. During the early stages of diapause, insects will expel free water from their bodies, reduce water content, and increase body fluid concentration to prevent cold damage [57]. Previous studies have reported this phenomenon from diapausing *Orius sauteri*, *Leguminivora glycinivorella*, and *Grapholita molesta* [58,59,60].

Lipids are important energy storage substances for diapausing insects such as *Drosophila suzukii* and *Chrysopa formosa* Brauer [61,62], and their levels vary during diapause. The main storage form is triglycerides, which usually account for 80% to 95% of total fat. Our results showed that triglyceride content was significantly higher in *A. aphidimyza* on diapause day 15 than during non-diapause. This indicated that *A. aphidimyza* accumulates a large amount of fat as its energy supply to ensure successful survival during the diapausing stage. With the extension of diapause time, the triglyceride content gradually decreases, possibly due to energy metabolism during diapause, consuming a portion of fat or converting fat into other energy substances for storage [63]. Lipids and triglycerides showed different patterns in content as diapause was induced, which may be due to lipids including not only triglycerides but also many other types. The results of this study showed that the significant increase in triglyceride content is due to the need to accumulate a large number of triglycerides to maintain one’s own needs after entering diapause, whereas the total lipid content showed a downward trend but no significant difference, which may be due to the high consumption of other lipids, leading to a decrease in total lipid content. Insects such as *Aspongopus chinensis*, *C. sinica* Tjeder, and *Kallima inachus* have also been reported to exhibit dynamic changes in fat accumulation during the early stages of diapause, with a gradual decrease in triglyceride content with the extension of diapause time [64,65,66].

Trehalose is a non-reducing disaccharide and the primary sugar found in the hemolymph of insects. It can influence carbohydrate metabolism, balance nutrients, enhance cold resistance, provide energy, resist stress responses, and induce chitin metabolism in diapause insects [43,67]. Our results showed that the trehalose content at diapause in *A. aphidimyza* was slightly higher than at non-diapause. This may be because *A. aphidimyza* accumulated trehalose to improve its cold tolerance. This is consistent with the rapid accumulation of trehalose in diapausing *Cydia pomonella* than at non-diapause when subjected to low-temperature stress [68,69]. Glycogen is an important energy storage substance in diapausing insects, providing nutrients for their life activities. Our results showed that the glycogen content in *A. aphidimyza* significantly increased on D15 but decreased on D25. This may be because *A. aphidimyza* does not feed after entering diapause and, therefore, converts its glycogen into glucose for energy and synthesizes antifreeze agents such as polyols to improve cold tolerance and maintain physiological activities [58,59,60]. The results showed that glycogen content increased on D35. This indicates that, at diapause, trehalose may have been converted into glycogen to provide energy during diapause and for growth and development after diapause. Trehalose and glycogen conversion have also been observed during diapause in *Antherea pernyi* [70]. Also, during the initial stage of diapause, in the silkworm [39] and *Ceropastes japonicus* [71], the algal sugar significantly decreases converts into glycogen, and is stored in the insect body, providing energy for the insects.

Sorbitol is also a cryoprotectant that can improve antifreeze resistance, promote stable osmotic pressure in the body, and further stabilize insects at low temperatures, playing an important role during insect diapause. Our results showed that sorbitol content was slightly higher in *A. aphidimyza* on D15 than in ND and further increased on D25 and D35 to improve the ability to resist stress. Similar results showing increased cold tolerance with an increase in sorbitol, glycogen, and trehalose content have been reported for *Lymantria dispar*, *Sitodiplosis mosellana* Gehin, *Artemia sinica*, *Pieris brassicae*, and *Apis mellifera* ligustica [72,73,74].

Our study showed that, during diapause in *A. aphidimyza*, these insects accumulate antifreeze protectants such as trehalose and sorbitol and energy storage substances such as glycogen and triglycerides to improve their adaptability to low temperatures and survival rates, promoting their population.

## Figures and Tables

**Figure 1 insects-15-00299-f001:**
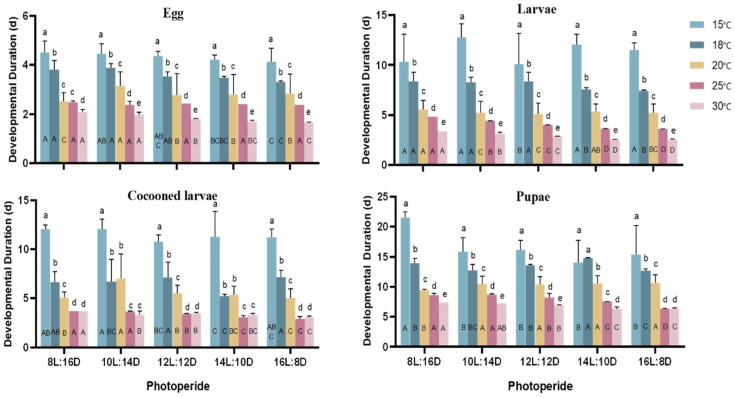
Effects of different temperatures and photoperiods on the developmental duration of *A. aphidimyza*. Capital letters indicate the impact of different photoperiods on the developmental periods, while lowercase letters indicate the significant differences in the effects of different temperatures on the developmental periods of *A. aphidimyza* after Tukey testing. (Only non-diapausing larvae were included in the estimation of development duration).

**Figure 2 insects-15-00299-f002:**
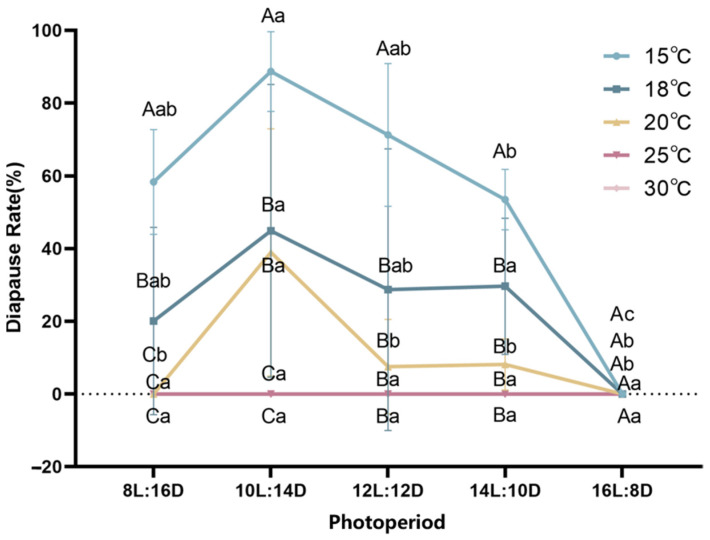
Effects of temperature and photoperiod on diapause in *A. aphidimyza*. Capital letters indicate the impact of different temperatures on the diapause of *A. aphidimyza*, and lowercase letters indicate the impact of different photoperiods on the diapause of *A. aphidimyza*. (Notes: Diapause rates at 25 °C and 30 °C are both 0).

**Figure 3 insects-15-00299-f003:**
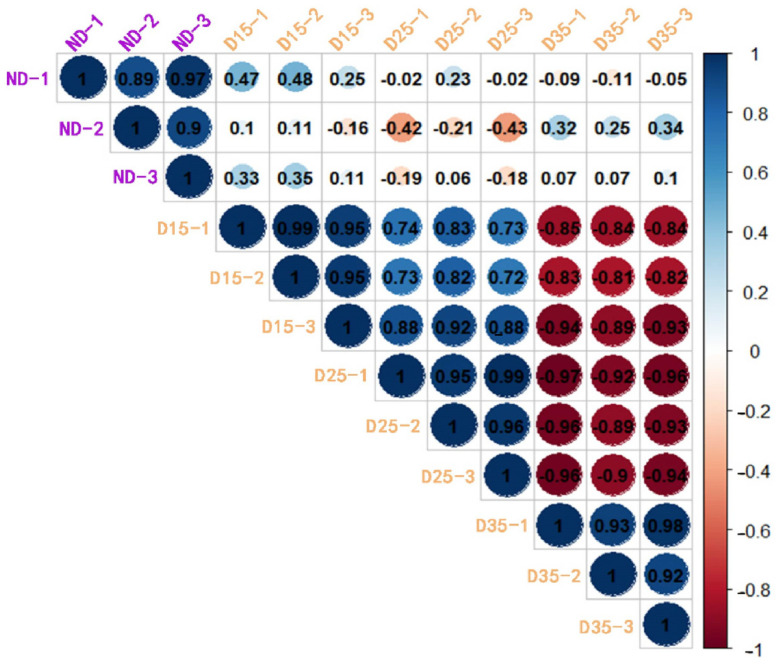
Repeatability analysis using Pearson hierarchical clustering.

**Figure 4 insects-15-00299-f004:**
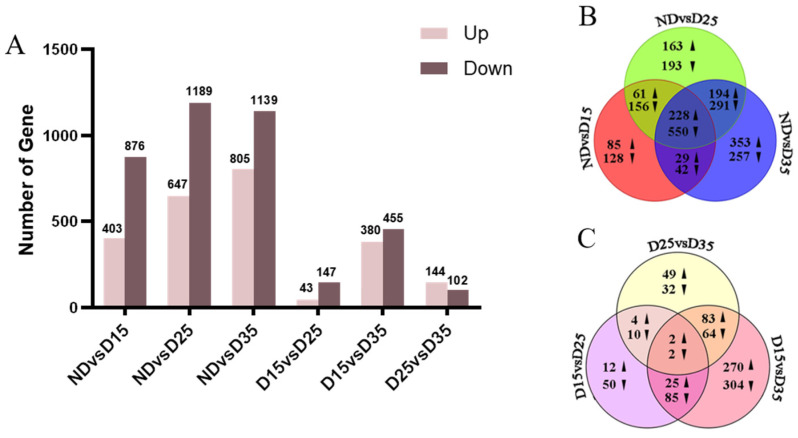
Column chart and Venn diagram of the numbers of differentially expressed genes. (**A**) differentially expressed genes; (**B**,**C**) Venn diagram of differentially expressed genes at different diapause stages.

**Figure 5 insects-15-00299-f005:**
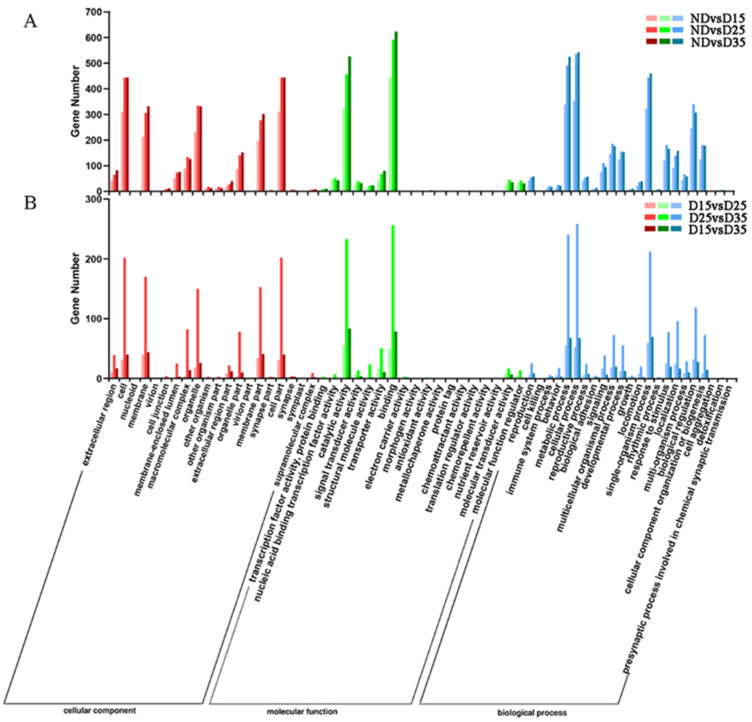
GO enrichment analysis of differentially expressed genes at different diapause phases. (**A**) Comparison between non-diapause stage and three diapause stages; (**B**) comparison of three diapause stages.

**Figure 6 insects-15-00299-f006:**
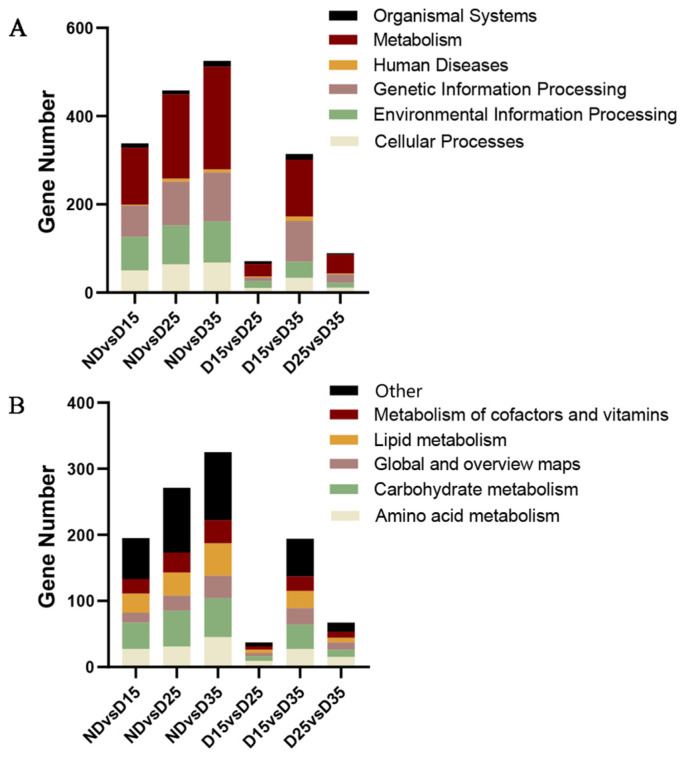
Distribution of enriched pathways of differentially expressed genes: (**A**) primary metabolic pathway; (**B**) metabolic pathway.

**Figure 7 insects-15-00299-f007:**
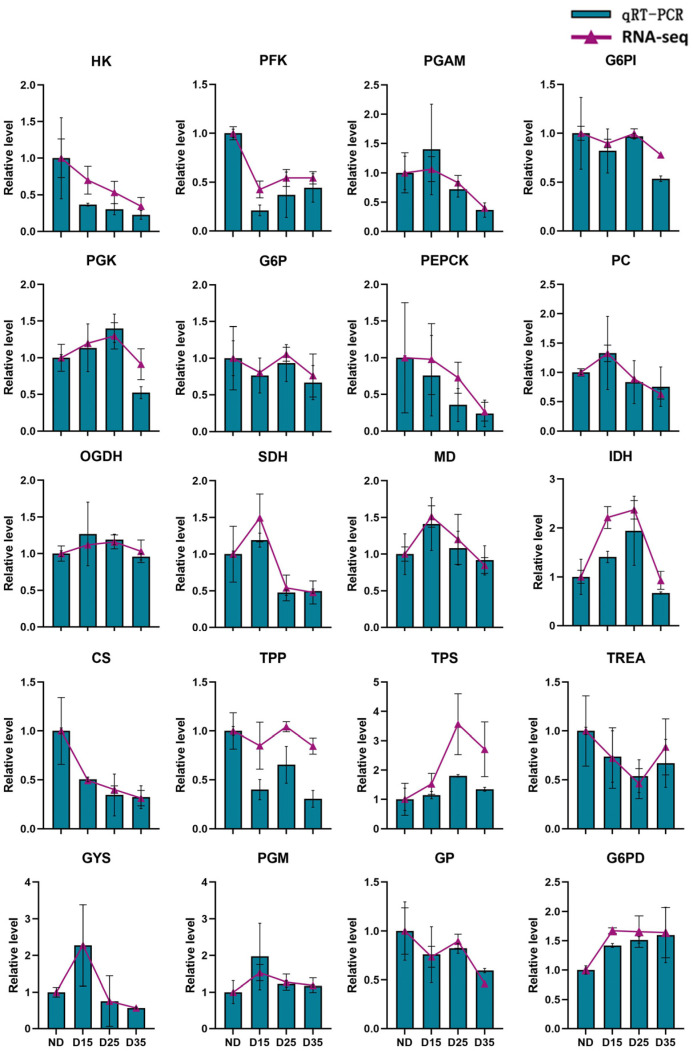
Validation of expressions of different genes by qRT-PCR.

**Figure 8 insects-15-00299-f008:**
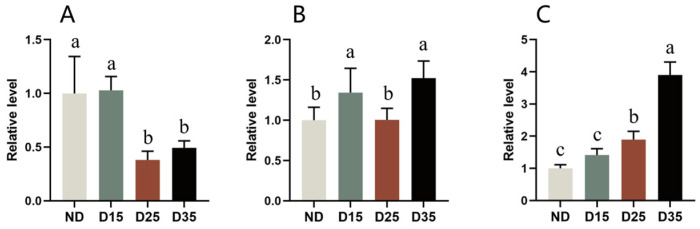
Changes in water (**A**), lipid (**B**), and triglyceride (**C**) contents in *A. aphidimyza* at different diapause stages. Different letters within columns (a, b, and c) are significantly different based on Tukey’s test (*p* < 0.05) (same as below).

**Figure 9 insects-15-00299-f009:**
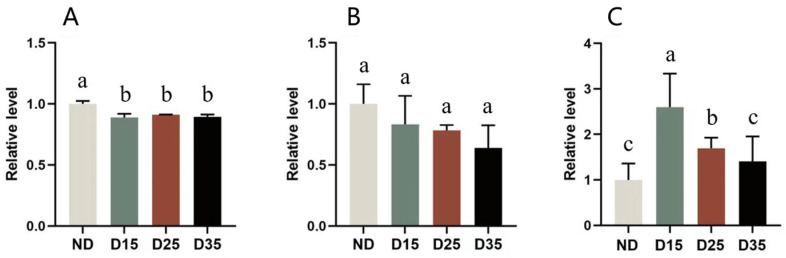
Changes in trehalose (**A**), glycogen (**B**), and sorbitol (**C**) contents in *A. aphidimyza* at different diapause stages.

**Table 1 insects-15-00299-t001:** The diapausing period of *A. aphidimyza* cocooned larvae.

Diapause Standard	Development Duration of Cocooned Larvae
non-diapause	≤5 d
incomplete diapause	6–15 d
complete diapause	>15 d

**Table 2 insects-15-00299-t002:** Developmental stages of *A. aphidimyza* at different photoperiods of 10L:14D (S) and 16L:8D (L) at 15 °C.

Treatment	Different Developmental Stages				
	Egg	1st Instar Larva	2nd Instar Larva	3rd Instar Larva	Cocooned Larvae
1	S	L	L	L	L
2	S	S	L	L	L
3	S	S	S	L	L
4	S	S	S	S	L
5	S	S	S	S	S
6	L	S	S	S	S
7	L	L	S	S	S
8	L	L	L	S	S
9	L	L	L	L	S
10	L	L	L	L	L

**Table 3 insects-15-00299-t003:** Summary of ANOVA results on the effects of different temperatures and photoperiods on the developmental duration of *A. aphidimyza*.

Biological Parameters	Source	df	F	*p*
Egg	Temperatures	4	1207.353	<0.05
	Photoperiods	4	18.181	<0.05
	Temperatures × Photoperiods	16	8.168	<0.05
	Error	3811		
Larvae	Temperatures	4	6648.596	<0.05
	Photoperiods	4	92.349	<0.05
	Temperatures × Photoperiods	16	28.389	<0.05
	Error	3876		
Cocooned larvae	Temperatures	4	2131.563	<0.05
	Photoperiods	4	26.552	<0.05
	Temperatures × Photoperiods	16	16.991	<0.05
	Error	3721		
Pupae	Temperatures	4	2089.923	<0.05
	Photoperiods	4	102.040	<0.05
	Temperatures × Photoperiods	16	45.208	<0.05
	Error	3721		

**Table 4 insects-15-00299-t004:** The diapause rate under photoperiods of 10L:14D (S) and 16L:8D (L) at different developmental stages in *A. aphidimyza* at 15 °C. Treatment details are described in Materials and Methods (Table 2). Notes: Values (means ± SE) followed by different letters within columns are significantly different based on Tukey’s test (*p* < 0.05).

Treatment	Diapause Rate (%)
1	16.25 ± 5.70 ^c^
2	59.38 ± 12.04 ^b^
3	83.33 ± 4.41 ^ab^
4	100.00 ± 0.00 ^a^
5	100.00 ± 0.00 ^a^
6	83.25 ± 3.98 ^ab^
7	17.84 ± 2.60 ^c^
8	16.78 ± 3.51 ^c^
9	13.81 ± 6.92 ^c^
10	0.00 ± 0.00 ^c^

## Data Availability

The datasets presented in this study can be found in online repositories. The names of the repository/repositories and accession number(s) can be found at: https://www.ncbi.nlm.nih.gov/sra/PRJNA1051789 (accessed on 12 December 2023.)

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
