# Peer review of "Molecular Correlates of Diapause in Aphidoletes aphidimyza"

_insects, 2024, doi:10.3390/insects15050299_

Round 1

Reviewer 1 Report

Comments and Suggestions for Authors

In this study, the authors investigated 1effects of photoperiod and temperature on diapause induction, 2) sensitive developmental stages for diapause induction, 3) diapause-associated gene expression and biochemical changes  in A. aphidimyza, a dominant natural enemy of aphids. The authors were able to obtain a lot of data on diapause of this species. These findings will be helpful for artificial or commercial breeding of this biocontrol agent. 

Overall the paper is an interesting, but I found this manuscript includes several points that should be considered by the authors. 

This paper contains a lot of data on diapause in this species. That's fine, but I feel that the focus of the research is becoming a little unclear. What is the real goal of this research? Presumably, it is to collect basic data on diapause to promote commercial breeding or in-depth research on molecular mechanisms underlying diapause regulation of this insect as you mentioned at the end of abstract. If so, I think it would be better to to point out how these results will be useful for mass production of this species near future and for molecular biological research on diapause of this species at the end of discussion. Please consider it. 

The vertical axes in Figures 8 and 9 are “relative expressions”, but what does this mean and how was this value determined? Please write the definition and calculation method on M&M.

 In the references, the bibliography contains a mixture of official journal names and abbreviations. I think it would be better to unify one or the other.

(Minor point) 

Line 65 Hevelka (1980)   Please include the paper number listed in the references.

Line 67 Masafumi et al (2012)  Please include the paper number.

Line 80 Yocum et al (2011)   Please include the paper number.

Line 103 Aphis craccivora à A. craccivora

Line 105 14L: 10D à 14 hour light and 10 hour dark (14L: 10D).  This notation is difficult for non-specialist readers to understand.

Line 148 ・・physiological adaptation of A. aphidimyza to diapause ・・ 

It is difficult to understand what is meant by the sentence physiological adaptation to torpor. What does it mean to adapt to diapause?

Line 213  A. aphidimyza à A. aphidimyza

Line 393-394  the developmental stages of Epicauta gorhami à the developmental duration of Epicauta gorhami

Line 411 Zhou et al. (2018) and Wu (2013)  Please include the paper number.

Line 420-26 How do your results differ from those of other studies? Please describe the differences in more details.

Line 526  I can not understand the meaning of “the accumulation of trehalose biosynthesis”.

Accumulation of trehalose or trehalose biosynthesis (?)

Line 526 promoted diapause à promoted by diapause (?)

Line 551  You mentioned that “The main storage form is triglycerides, which usually account for 80% to 95% of total fat”. But as shown in Fig. 8, lipids and triglycerides showed different patterns in content as diapause was induced. What does this difference mean? I think some explanation is needed regarding this.

Line641 0xygen à Oxygen

Author Response

Response to Reviewer 1

In this study, the authors investigated 1)effects of photoperiod and temperature on diapause induction, 2) sensitive developmental stages for diapause induction, 3) diapause-associated gene expression and biochemical changes  in A. aphidimyza, a dominant natural enemy of aphids. The authors were able to obtain a lot of data on diapause of this species. These findings will be helpful for artificial or commercial breeding of this biocontrol agent. Overall the paper is an interesting, but I found this manuscript includes several points that should be considered by the authors. 

・This paper contains a lot of data on diapause in this species. That's fine, but I feel that the focus of the research is becoming a little unclear. What is the real goal of this research? Presumably, it is to collect basic data on diapause to promote commercial breeding or in-depth research on molecular mechanisms underlying diapause regulation of this insect as you mentioned at the end of abstract. If so, I think it would be better to point out how these results will be useful for mass production of this species near future and for molecular biological research on diapause of this species at the end of discussion. Please consider it. 

Response: Thank you for the reviewer’s advice. We have added the contents in the discussion. “This study successfully induced diapause in A. aphidimyza, elucidating the tempera-ture and photoperiod requirements for diapause induction in this mosquito species. These findings provide technical support and theoretical guidance for the artificial control of diapause, extending the shelf life of products, and enhancing the effectiveness of applica-tion in pest control. This research is of significant importance for the commercial produc-tion and large-scale application of A. aphidimyza.” See L460-465.

・The vertical axes in Figures 8 and 9 are “relative expressions”, but what does this mean and how was this value determined? Please write the definition and calculation method on M&M.

Response: Thank you for the reviewer’s advice. We have added the contents in the discussion. “The relative level = 2-ΔΔCT and means that the expression of the target gene is calculated by comparing the target gene with the internal reference gene.” See L167-168.

 ・In the references, the bibliography contains a mixture of official journal names and abbreviations. I think it would be better to unify one or the other.

Response: Thank you for the reviewer’s advice. We have changed the references. See the references.

 ・Line 65 Hevelka (1980) Please include the paper number listed in the references.

Response: Thank you for the reviewer’s advice. We have added the paper number “[14]”. See L77.

 ・Line 67 Masafumi et al (2012)  Please include the paper number.

Response: Thank you for the reviewer’s advice. We have added the paper number “[22]”. See L79.

 ・Line 80 Yocum et al (2011)   Please include the paper number.

Response: Thank you for the reviewer’s advice. We have added the paper number “[9]”. See L92

 ・Line 103 Aphis craccivora à A. craccivora

Response: Thank you for the reviewer’s advice. We have changed the Aphis craccivora to A. craccivora. See L113

 ・Line 105 14L: 10D à 14 hour light and 10 hour dark (14L: 10D).  This notation is difficult for non-specialist readers to understand.

Response: Thank you for the reviewer’s advice. We have changed the 14L: 10D to 14 hour light and 10 hour dark (14L: 10D). See L110 and L117-124

 ・Line 148 physiological adaptation of A. aphidimyza to diapause. It is difficult to understand what is meant by the sentence physiological adaptation to torpor. What does it mean to adapt to diapause?

Response: Thank you for the reviewer’s advice. This question due to adapt to diapause environment, some physiological characteristics in the body of A. aphidimyza will change, so it is called physiological adaptation.

 ・Line 213  A. aphidimyza à A. aphidimyza

Response: Thank you for the reviewer’s advice. We have changed the A. aphidimyza to A. aphidimyza all artical.

 ・Line 393-394  the developmental stages of Epicauta gorhami à the developmental duration of Epicauta gorhami

Response: Thank you for the reviewer’s advice. We have changed the developmental stages of Epicauta gorhami to “the developmental duration of Epicauta gorhami”. See L454

 ・Line 411 Zhou et al. (2018) and Wu (2013)  Please include the paper number.

Response: Thank you for the reviewer’s advice. We have inclued the paper number “[22]”. See L471

 ・Line 420-26 How do your results differ from those of other studies? Please describe the differences in more details.

Response: Thank you for the reviewer’s advice.We have added the contents in the discussion “Our results are different from other studies on diapause in A. aphidimyza [22,23], that the diapause condition is 15 ℃, 10L: 14D, while other reports that the diapause condition is 18 ℃, 8L: 16D, which may be related to geographical differences.”See L480-483

 ・Line 526  I can not understand the meaning of “the accumulation of trehalose biosynthesis”.

Accumulation of trehalose or trehalose biosynthesis (?)

Response: Thank you for the reviewer’s advice. We have changed the accumulation of trehalose biosynthesis to “the accumulation of trehalose . See L595

 ・Line 526 promoted diapause à promoted by diapause (?)

Response: Thank you for the reviewer’s advice. We have changed promoted diapause to promoted by diapause. See L595

 ・Line 551  You mentioned that “The main storage form is triglycerides, which usually account for 80% to 95% of total fat”. But as shown in Fig. 8, lipids and triglycerides showed different patterns in content as diapause was induced. What does this difference mean? I think some explanation is needed regarding this.

Response: Thank you for the reviewer’s advice.We have added the contents in the discussions “Lipids and triglycerides showed different patterns in content as diapause was induced, which may be due to lipids include not only triglycerides, but also many other types. The result of this study that the significant increase of triglyceride content is due to the need to accumulate a large amount of triglycerides to maintain one's own needs after entering diapause, while the total lipid content shows a downward trend, but no significant difference, which may be due to the high consumption of other lipids, leading to a decrease in total lipid content.” See L626-633

 ・Line641 0xygen à Oxygen

Response: Thank you for the reviewer’s advice. We have changed 0xygen to Oxygen. See L720

Reviewer 2 Report

Comments and Suggestions for Authors

The manuscript „The aspect of the molecular mechanisms underlying diapause in Aphidoletes aphidimyza“ submitted to Insects is based on a research which is generally well designed, conducted and analysed. The results are original and interesting, I have to appreciate an overwhelming amount of work done by the authors, especially need for rearing such numbers of insects in so many treatments. Thanks to this amount of work the results are probably very strong and definitely deserve publication. I have just several minor points that should be improved prior to publication. Next to the formal issues (see below), I think the authors should improve:

11)        Statistical analysis of the changes of body content and metabolites (presented at figs. 8 and 9) – the repeated measures ANOVA would be appropriate test as the subsequent samples are not independent.

22)        A background and discussion on the length of the development. Authors should at least mention (if not test) a concept of the developmental rate isomorphy (see e.g. Jarosik et al. 2002; Kuang et al. 2012; Boukal et al. 2015)

The formal issues are relatively abundant throughout the manuscript, some parts have probably “escaped” from the final correction.

11)        The citation Kostal (2006) is not in the references (l. 63)

22)        The tables 2 and 4 are actually almost the same, I suggest remove the table 2

33)        The Table 3 referenced at l. 134 is definitely not the table 3 (at the same page)

44)        The data analyses are not complete, the authors mention only the post-hoc analysis

55)        The sentence “Changes in water content need more relevant citations” (l. 541) is probably not intended to be part of the manuscript.

After these corrections I think the manuscript can be accepted for publication in Insects. 

Comments on the Quality of English Language

The text should be revised carefully, there are many mistakes in interpunction, word order and articles, but I don't feel competent to revise the text as I am not a native speaker.

Author Response

Response to Reviewer 2

The manuscript, The aspect of the molecular mechanisms underlying diapause in Aphidoletes aphidimyza“ submitted to Insects is based on a research which is generally well designed, conducted and analysed. The results are original and interesting, I have to appreciate an overwhelming amount of work done by the authors, especially need for rearing such numbers of insects in so many treatments. Thanks to this amount of work the results are probably very strong and definitely deserve publication. I have just several minor points that should be improved prior to publication. Next to the formal issues (see below), I think the authors should improve:

  • Statistical analysis of the changes of body content and metabolites (presented at figs. 8 and 9) – the repeated measures ANOVA would be appropriate test as the subsequent samples are not independent.

Response: Thank you for the reviewer’s advice. The statistical analyses in the paper are using one-way ANOVA, without using post-hoc analysis. See L178.

22)  A background and discussion on the length of the development. Authors should at least mention (if not test) a concept of the developmental rate isomorphy (see e.g. Jarosik et al. 2002; Kuang et al. 2012; Boukal et al. 2015)

Response: Thank you for the reviewer’s advice. We have added the development duration on background. “Development duration, temperature and photoperiod have always been important parts of insect ecology. Different temperature and photoperiod have different effects during growth and development period in A. aphidimyza. Females lived for about 4-9 days, males lived for about 3-5 days bred at 25℃, adults can lay eggs the next day after mating, larvae hatch from the front of the egg and feed on aphids after 2-3 days, cocooned larvae prepare for pupation after 4-6 days, and the pupal stage is maintained for 8-12 days. Lin Qingcai research showed that with the increase of temperature, the development rate of eggs and larvae of A. aphidimyza accelerated. Gong Yajun (2007) found that the optimal development temperature of A. aphidimyza was 19-23 ℃, and development was significantly inhibited at 31 ℃. Therefore, the study of the effect of temperature and photoperiod on the developmental period of insects is of great significance to understand the habits of insects and the release of natural enemies in the field.” See L60-71

  • The citation Kostal (2006) is not in the references (l. 63)

Response: Thank you for the reviewer’s advice. We have added this references. See L75.

  • The tables 2 and 4 are actually almost the same, I suggest remove the table 2

Response: Thank you for the reviewer’s advice. The table 2 is materials and methods, but table 4 is result, so we think do not remove these tables.

  • The Table 3 referenced at l. 134 is definitely not the table 3 (at the same page)

Response: Thank you for the reviewer’s advice. We have changed the table 3 to suitable location. See L216

  • The data analyses are not complete, the authors mention only the post-hoc analysis

Response: Thank you for the reviewer’s advice. The statistical analyses in the paper are using one-way ANOVA, without using post-hoc analysis. And we have modified in the data analysis “All statistical analyses were conducted using IBM SPSS statistics 23.0, and using the Tukey's test of variance, the results were expressed as mean ± standard error. Charts were created using GraphPad Prism 8.0.1.”. See L178-181

  • The sentence “Changes in water content need more relevant citations” (l. 541) is probably not intended to be part of the manuscript.

Response: Thank you for the reviewer’s advice. We have deleted the sentence “Changes in water content need more relevant citations”.

Reviewer 3 Report

Comments and Suggestions for Authors

Brief Summary:

The manuscript “The aspect of the molecular mechanisms underlying diapause in Aphidoletes aphidimyza” by Dai et al. presents their findings on the characterization of diapause and its molecular correlates in the midge A. aphidimyza. The authors studied the effects of photoperiod and temperature on diapause incidence in the midge A. aphidimyza and mapped the photoperiod sensitive phase. Further, they performed transcriptome sequencing and metabolic analysis on the diapausing and non-diapausing insect samples to gain insights into molecular correlates of diapause state. De novo assembly and annotation of A. aphidimyza transcriptome in this study has produced probably the first genomic resource in this species which will accelerate the study of molecular mechanisms of diapause regulation. (If it is indeed the first genomic/transcriptomic study in this species, I urge authors to emphasize on this point).

General comments:

1.       This study characterized effects of temperature and photoperiod on the diapause expression, followed by studying the molecular correlates of diapause state. The study does not go further and test the causal role of the candidate genes or metabolites in the development of diapause state. Therefore, “Molecular correlates of diapause in Aphidoletes aphidimyza” may be a more appropriate title of this manuscript.

2.       Diapause phenotyping methods have not been described in adequate details in this manuscript. More details may help explain many apparent discrepancies in the diapause data. Please see the specific comments for discrepancies or lack of clarity in the diapause data.

3.       GO analysis is a very important part of the study. Description of the GO analysis appears very superficial. There is a huge scope to improve the presentation and description of the GO analysis.

Specific comments:

1.       Table 1 shows criteria (developmental stage duration thresholds) used in the study to identify diapause state. What was the basis for using these criteria? Please cite the article if those criteria are based on any previously published study.

2.       Distinguishing diapause from non-diapause state in A. aphidimyza is based on the duration of cocooned larval stage. This study used a fixed threshold for cocooned larvae duration (6-15 days: incomplete diapause; >15 days: diapause) to determine if the insect is in diapause or not. The rate of development is highly sensitive to temperature. Typically, low ambient temperatures slow down the rate of development thereby lengthening the duration of all the developmental stages, but slow rate of development need not necessarily induce diapause (developmental arrest). I wonder if a fixed developmental stage duration threshold can be used for a range of temperatures used in this study?

3.       Please clarify the meaning of incomplete diapause. Is it a real diapause or a non-diapause state with just a slower rate of development? Was the term ‘incomplete diapause’ used for A. aphidimyza in any previously published study? If yes, please cite the article.

4.       Please provide the details of the method used for diapause incidence estimations. How do cocooned larval stage durations in Figure 1 translate into the diapause incidence estimates in Figure 2? Were the criteria showed in the Table 1 used to estimate diapause incidence? Cocooned larval stage duration did not exceed 15 days (Figure 1) in any light schedule at any temperature, does it mean the diapause reported in Figure 2 is all incomplete diapause (cocooned larval duration: 6-15 days)?

5.       Average cocooned larval stage duration under 16L:8D at 15oC (Figure 1) is about 12 days. It is not clear how does 12 days average cocooned larval stage duration translate into diapause rate of 0% (Figure 2)?

6.       Transcriptome analysis used the larvae that were in diapause for 15, 25 and 35 days (D15, D25 and D35) under 10L:14D. If there were individuals that remained in cocooned larval stage for 15, 25 and 35 days, why was the average duration of cocooned larval stage only 12-13 days under 10L:14D (Figure 1)? Was the percentage of larvae showing cocooned larval stage duration longer than 15 days very low? Please provide the percentage of larvae that showed cocooned stage longer than 15 days.

7.       Description of results about differential gene expression is confusing. While comparing genes up- or down- regulated between two groups, say A vs B, please clearly state if the genes are up- or down- regulated (1) in group A compared to B, or (2) in group B compared to A? for example in Figure 4 – ND vs D15, were the 403 genes upregulated in ND or in D15?

8.       Please provide more details about the methods used for estimation of metabolites. Which tissue samples were used for the estimation of metabolite levels – whole body or hemolymph samples? Were the metabolite levels normalized by total protein or weight of the sample to make the estimates comparable across treatments (ND vs D15 vs D25 vs D35)? Y-axis labels for Figure 8 and 9 are ‘relative expression’. For metabolites- ‘level’ is more appropriate word than ‘expression’, please replace ‘expression’ by ‘level’.  Please consider using the format e.g. relative level of lipids (%)? Please mention the units of metabolite levels for Figure 8 , 9 in the captions.

9.       Line 420-423: “Our results are different from other studies 420 on diapause in A. aphidimyza [22,23], which may be related to differences in geographical 421 locations. The critical length of diapause in A. aphidimyza was reported to vary at different 422 latitudes and decreases with decreasing latitude [8].” Please provide details about the A. aphidimyza strain such as source location, latitude, year of collection, rearing conditions etc. and if it explains the critical photoperiod observed in the study.

GO analysis description appears superficial and cluttered. GO terms in Figure 5 are not legible. Separate GO enrichment analysis for up- and down- regulated g

Comments on the Quality of English Language

Minor corrections in the phrasing and sentence construction may improve the readability.

Author Response

Response to Reviewer 3

The manuscript “The aspect of the molecular mechanisms underlying diapause in Aphidoletes aphidimyza” by Dai et al. presents their findings on the characterization of diapause and its molecular correlates in the midge A. aphidimyza. The authors studied the effects of photoperiod and temperature on diapause incidence in the midge A. aphidimyza and mapped the photoperiod sensitive phase. Further, they performed transcriptome sequencing and metabolic analysis on the diapausing and non-diapausing insect samples to gain insights into molecular correlates of diapause state. De novo assembly and annotation of A. aphidimyza transcriptome in this study has produced probably the first genomic resource in this species which will accelerate the study of molecular mechanisms of diapause regulation. (If it is indeed the first genomic/transcriptomic study in this species, I urge authors to emphasize on this point).

General comments:

  1. This study characterized effects of temperature and photoperiod on the diapause expression, followed by studying the molecular correlates of diapause state. The study does not go further and test the causal role of the candidate genes or metabolites in the development of diapause state. Therefore, “Molecular correlates of diapause in Aphidoletes aphidimyza” may be a more appropriate title of this manuscript.

Response: we have modified the title as “Molecular correlates of diapause in Aphidoletes aphidimyza”. See L1.

  1. Diapause phenotyping methods have not been described in adequate details in this manuscript. More details may help explain many apparent discrepancies in the diapause data. Please see the specific comments for discrepancies or lack of clarity in the diapause data.

Response: we have see the specific comments for discrepancies or lack of clarity in the diapause data. And have modified the methods.

  1. GO analysis is a very important part of the study. Description of the GO analysis appears very superficial. There is a huge scope to improve the presentation and description of the GO analysis.

Response: we have added some Go analysis. “GO functional annotation was performed on the differentially expressed genes ob-tained by pairwise comparison between the non-diapause group and three different dia-pause groups at different times (Figure 5). The results showed that ND vs D15d, ND vs D25d, ND vs D35d, D15d vs D25d, D15d vs D35d, D25d vs D35d, and D25d vs D35d each annotated 809; 1,129; 1,198; 113; 545 and 151 differentially expressed genes enriched into the three major categories of GO functions, mainly including biological processes, cell composition, and molecular functions. Pairwise comparison at each stage showed that the differentially expressed genes were mainly enriched in the Cell, Membrane, and Organelle of the cell composition; Cellular process, Metabolic pro-cess, and Single-organism process were more abundant in biological processes; Binding, Catalytic activity, and Transporter activity were more abundant in molecular functions.” See L351-355.

Specific comments:

  1. Table 1 shows criteria (developmental stage duration thresholds) used in the study to identify diapause state. What was the basis for using these criteria? Please cite the article if those criteria are based on any previously published study.

Response: Thank you for the reviewer’s advice. We have added the reference about diapause state “[28]Zhai Y.F, Zhang X.Y, Yin Z.J, Zhu Q.S, Tao M, Yu Y, Zheng L. Nymphal Diapause in Laodelphax striatellus (Hemiptera: Delphacidae). Journal of Entomological Science. 2018,53(2):107-122. http://doi.org/10.18474/JES17-140.1 ” See L135

  1. Distinguishing diapause from non-diapause state in  aphidimyzais based on the duration of cocooned larval stage. This study used a fixed threshold for cocooned larvae duration (6-15 days: incomplete diapause; >15 days: diapause) to determine if the insect is in diapause or not. The rate of development is highly sensitive to temperature. Typically, low ambient temperatures slow down the rate of development thereby lengthening the duration of all the developmental stages, but slow rate of development need not necessarily induce diapause (developmental arrest). I wonder if a fixed developmental stage duration threshold can be used for a range of temperatures used in this study?

Response: Thank you for the reviewer’s advice. We did a lot of preliminary experiments on the division of diapause state in the early stage. We have found that the time for the cocooned larvae to enter the pupal stage did not exceed 15 days no matter what photoperiod conditions, and the cocooned larvae remained in the stage beyond 15 days. Moreover, the development duration of the cocooned larvae was 3-5 days at normal conditions. Therefore this period is used as the critical point for dividing diapause state.

3.Please clarify the meaning of incomplete diapause. Is it a real diapause or a non-diapause state with just a slower rate of development? Was the term ‘incomplete diapause’ used for A. aphidimyza in any previously published study? If yes, please cite the article.

Response: Thank you for the reviewer’s advice. Incomplete diapause refers to the period when the cocooned larvae develops to the pupal stage, because the cocooned larvae is composed of diapause cocooned larvae and non-diapause cocooned larvae at this time, so it is defined as incomplete diapause.

  1. Please provide the details of the method used for diapause incidence estimations.

How do cocooned larval stage durations in Figure 1 translate into the diapause incidence estimates in Figure 2? Were the criteria showed in the Table 1 used to estimate diapause incidence? Cocooned larval stage duration did not exceed 15 days (Figure 1) in any light schedule at any temperature, does it mean the diapause reported in Figure 2 is all incomplete diapause (cocooned larval duration: 6-15 days)?

Response: Thank you for the reviewer’s advice. Diapause incidence = Number of insects that have not entered the pupal stage for more than 15 days/The number of insects tested. The diapause state are used to calculate diapause incidence in Table 1. The diapause was the incidence of complete diapause in Figure 2, with cocooned larvae for more than 15 days.

  1. Average cocooned larval stage duration under 16L:8D at 15oC (Figure 1) is about 12 days. It is not clear how does 12 days average cocooned larval stage duration translate into diapause rate of 0% (Figure 2)?

Response: Thank you for the reviewer’s advice. The diapause rate of cocooned larvae was 0% under 16L:8D at 15℃, because the cocooned larvae entered the pupal stage in an average time of 12 days, and did not maintain the status of cocooned larvae.

  1. Transcriptome analysis used the larvae that were in diapause for 15, 25 and 35 days (D15, D25 and D35) under 10L:14D. If there were individuals that remained in cocooned larval stage for 15, 25 and 35 days, why was the average duration of cocooned larval stage only 12-13 days under 10L:14D (Figure 1)? Was the percentage of larvae showing cocooned larval stage duration longer than 15 days very low? Please provide the percentage of larvae that showed cocooned stage longer than 15 days.

Response: Thank you for the reviewer’s advice. The average duration of cocooned larvae stage only 12-13 days is a synthesis of all the data. Some of the insects used in the test entered the pupal stage at 15 days, and most of the non-diapause state entered the pupal stage at 12-13 days, so 12-13 days are shown in Figure 1. When A. aphidimyza enter diapause, they maintain the form of cocooned larvae beyond 15 days, and the proportion of cocooned larvae under 10L:14D photoperiod condition is very high, as can be seen in the diapause rate under 10:14 photoperiod condition in Figure 2 and the diapause rate was as high as 88.7%.

  1. Description of results about differential gene expression is confusing. While comparing genes up- or down- regulated between two groups, say A vs B, please clearly state if the genes are up- or down- regulated (1) in group A compared to B, or (2) in group B compared to A? for example in Figure 4 – ND vs D15, were the 403 genes upregulated in ND or in D15?

Response: Thank you for the reviewer’s advice. We have modified the comparison of gene expression in the article.  “The results showed that 1,279 genes showed differential expressed in D15 compared with ND, of which 403 genes were up-regulated and 876 genes were down-regulated. In D25 compared with ND, 1,836 genes showed differential expression, of which 647 were up-regulated and 1,189 were down-regulated. In D35 compared with ND, 1,944 genes showed differential expression, of which 805 were up-regulated and 1,139 were down-regulated. In D25 compared with D15, 190 genes showed differential expression, of which 43 genes were up-regulated and 147 genes were down-regulated. In D35 compared with D15, 835 genes showed differential expression, of which 380 genes were up-regulated and 455 genes were down-regulated. In D35 compared with D25, 246 genes showed differential expression, of which 144 genes were up-regulated and 102 genes were down-regulated. The down-regulated differentially expressed genes were higher than the up-regulated differentially expressed genes in all other treatment groups, except for D25 vs D35.

A total of 2,370 differentially expressed genes were compared among the ND vs D15, ND vs D25, and ND vs D35 groups (Figure 4B). Among them, 778 differentially expressed genes were co-expressed between groups, of these 228 genes were upregulated, and 550 genes were downregulated. There are 213 differentially expressed genes specifically expressed in D15 compared with ND, of which 85 genes were up-regulated and 128 genes were down-regulated. In D25 compared with ND comparison group, there were 356 differentially expressed genes, of which 163 genes were up-regulated and 193 genes were down-regulated; In D35 compared with ND, there were 610 differentially expressed genes, of which 353 genes were up-regulated and 257 genes were down-regulated.

A total of 992 differentially expressed genes were compared among the D15 vs D25, D15 vs D35, and D25 vs D35 groups (Figure 4C). Among them, 4 differentially expressed genes were co-expressed between groups, with 2 genes up-regulated and 2 genes down-regulated. There were 62 differentially expressed genes in D25 compared with D15, of which 12 genes are up-regulated and 50 genes are down-regulated. There were 574 differentially expressed genes in D35 compared with D15, of which 270 genes were up-regulated and 304 genes were down-regulated; 81 differentially expressed genes were specifically expressed in D35 compared with D25, with 49 genes up-regulated and 32 genes down-regulated.” See L284-343

  1. Please provide more details about the methods used for estimation of metabolites. Which tissue samples were used for the estimation of metabolite levels – whole body or hemolymph samples? Were the metabolite levels normalized by total protein or weight of the sample to make the estimates comparable across treatments (ND vs D15 vs D25 vs D35)? Y-axis labels for Figure 8 and 9 are ‘relative expression’. For metabolites- ‘level’ is more appropriate word than ‘expression’, please replace ‘expression’ by ‘level’.  Please consider using the format e.g. relative level of lipids (%)? Please mention the units of metabolite levels for Figure 8 , 9 in the captions.

Response: Thank you for the reviewer’s advice. The methods used for estimation of metabolites was used whole body and normalized according to total protein, we have modified ‘expression’ to ‘level’ in the article.

  1. Line 420-423: “Our results are different from other studies 420 on diapause in  aphidimyza [22,23], which may be related to differences in geographical 421 locations. The critical length of diapause in A. aphidimyza was reported to vary at different 422 latitudes and decreases with decreasing latitude [8].” Please provide details about the A. aphidimyza strain such as source location, latitude, year of collection, rearing conditions etc. and if it explains the critical photoperiod observed in the study. GO analysis description appears superficial and cluttered. GO terms in Figure 5 are not legible. Separate GO enrichment analysis for up- and down- regulated g

Response: Thank you for the reviewer’s advice. We have supplemented in the material and methods “The tested insects, A. aphidimyza, A. craccivora, were reared in glass greenhouse in Institute of Plant Protection, Shandong Academy of Agricultural Sciences(Shandong, China, N 36°41’, E 116°54’). Broad bean seedlings were used as host plants to breed Aphis craccivora, which was then used to breed A. aphidimyza. All insect-rearing procedures were performed at 25 ℃, relative humidity of 65%~75%, and photoperiod of 14 hour light and 10 hour dark (14L: 10D).” See L114-124

Round 2

Reviewer 1 Report

Comments and Suggestions for Authors

The authors properly addressed the points that were raised in my review of the previous version.

Author Response

The authors properly addressed the points that were raised in my review of the previous version.

Response: Thank you for the reviewer’s valuable advice.

Reviewer 2 Report

Comments and Suggestions for Authors

The manuscript looks a bit improved, but the main problem is still not solved. The statistical test used for analysis of changes during diapause (sections 3.9 and 3.10) has to be a test for dependent samples, as the samples are not independent on each other (neither one-way ANOVA nor Student's t-test (Fig. 8 and fig. 9) can be used). I suggest Repeated-measures ANOVA (or e.g. Friedman test, but the Repeated-measures ANOVA has greater power) shall be used.

Again, I think that the tables 2 and 4 are actually almost the same, I suggest remove the table 2 - without removing you are actually elongating the manuscript without reason.

Author Response

Response to Reviewer 2 Comments

The manuscript looks a bit improved, but the main problem is still not solved. The statistical test used for analysis of changes during diapause (sections 3.9 and 3.10) has to be a test for dependent samples, as the samples are not independent on each other (neither one-way ANOVA nor Student's t-test (Fig. 8 and fig. 9) can be used). I suggest Repeated-measures ANOVA (or e.g. Friedman test, but the Repeated-measures ANOVA has greater power) shall be used.

Response: We have modified the Fig 8 and 9.

Figure 8. Changes in water (A), lipid (B) and triglyceride (C) contents in A. aphidimyza at different diapause stages. Different letters within columns (a, b, c) are significantly different by Tukey,s test (p < 0.05). (Same as the below).

Figure 9. Changes in of trehalose (A), glycogen (B) and sorbitol (C) contents in A. aphidimyza at different diapause stages.

Again, I think that the tables 2 and 4 are actually almost the same, I suggest remove the table 2 - without removing you are actually elongating the manuscript without reason.

Response: Thank you for the reviewer’s valuable advice. Table 2 showed the different light duration in the materials and methods, and Table 4 showed the diapause rate with different light duration, so it needs to be retained.

Reviewer 3 Report

Comments and Suggestions for Authors

Reviewer’s response to author revision

Authors have satisfactorily addressed some concerns, but remaining concerns need to be appropriately addressed. Please find my responses highlighted in red text.

The manuscript “The aspect of the molecular mechanisms underlying diapause in Aphidoletes aphidimyza” by Dai et al. presents their findings on the characterization of diapause and its molecular correlates in the midge A. aphidimyza. The authors studied the effects of photoperiod and temperature on diapause incidence in the midge A. aphidimyza and mapped the photoperiod sensitive phase. Further, they performed transcriptome sequencing and metabolic analysis on the diapausing and non-diapausing insect samples to gain insights into molecular correlates of diapause state. De novo assembly and annotation of A. aphidimyza transcriptome in this study has produced probably the first genomic resource in this species which will accelerate the study of molecular mechanisms of diapause regulation. (If it is indeed the first genomic/transcriptomic study in this species, I urge authors to emphasize on this point).

General comments:

This study characterized effects of temperature and photoperiod on the diapause expression, followed by studying the molecular correlates of diapause state. The study does not go further and test the causal role of the candidate genes or metabolites in the development of diapause state. Therefore, “Molecular correlates of diapause in Aphidoletes aphidimyza” may be a more appropriate title of this manuscript.

Response: we have modified the title as “Molecular correlates of diapause in Aphidoletes aphidimyza”. See L1.

>>Reviewer’s response: The word “correlates” is missing from the main title.

Diapause phenotyping methods have not been described in adequate details in this manuscript. More details may help explain many apparent discrepancies in the diapause data. Please see the specific comments for discrepancies or lack of clarity in the diapause data.

Response: we have see the specific comments for discrepancies or lack of clarity in the diapause data. And have modified the methods.

GO analysis is a very important part of the study. Description of the GO analysis appears very superficial. There is a huge scope to improve the presentation and description of the GO analysis.

Response: we have added some Go analysis. “GO functional annotation was performed on the differentially expressed genes ob-tained by pairwise comparison between the non-diapause group and three different dia-pause groups at different times (Figure 5). The results showed that ND vs D15d, ND vs D25d, ND vs D35d, D15d vs D25d, D15d vs D35d, D25d vs D35d, and D25d vs D35d each annotated 809; 1,129; 1,198; 113; 545 and 151 differentially expressed genes enriched into the three major categories of GO functions, mainly including biological processes, cell composition, and molecular functions. Pairwise comparison at each stage showed that the differentially expressed genes were mainly enriched in the Cell, Membrane, and Organelle of the cell composition; Cellular process, Metabolic pro-cess, and Single-organism process were more abundant in biological processes; Binding, Catalytic activity, and Transporter activity were more abundant in molecular functions.” See L351-355.

Specific comments:

Table 1 shows criteria (developmental stage duration thresholds) used in the study to identify diapause state. What was the basis for using these criteria? Please cite the article if those criteria are based on any previously published study.

Response: Thank you for the reviewer’s advice. We have added the reference about diapause state “[28]Zhai Y.F, Zhang X.Y, Yin Z.J, Zhu Q.S, Tao M, Yu Y, Zheng L. Nymphal Diapause in Laodelphax striatellus (Hemiptera: Delphacidae). Journal of Entomological Science. 2018,53(2):107-122. http://doi.org/10.18474/JES17-140.1 ” See L135

>>Reviewer’s response: Please cite a study in A. aphidimyza where such criteria have been used.

Distinguishing diapause from non-diapause state in aphidimyzais based on the duration of cocooned larval stage. This study used a fixed threshold for cocooned larvae duration (6-15 days: incomplete diapause; >15 days: diapause) to determine if the insect is in diapause or not. The rate of development is highly sensitive to temperature. Typically, low ambient temperatures slow down the rate of development thereby lengthening the duration of all the developmental stages, but slow rate of development need not necessarily induce diapause (developmental arrest). I wonder if a fixed developmental stage duration threshold can be used for a range of temperatures used in this study?

Response: Thank you for the reviewer’s advice. We did a lot of preliminary experiments on the division of diapause state in the early stage. We have found that the time for the cocooned larvae to enter the pupal stage did not exceed 15 days no matter what photoperiod conditions, and the cocooned larvae remained in the stage beyond 15 days. Moreover, the development duration of the cocooned larvae was 3-5 days at normal conditions. Therefore this period is used as the critical point for dividing diapause state.

>>Reviewer’s response: Addressed satisfactorily.

3.Please clarify the meaning of incomplete diapause. Is it a real diapause or a non-diapause state with just a slower rate of development? Was the term ‘incomplete diapause’ used for A. aphidimyza in any previously published study? If yes, please cite the article.

Response: Thank you for the reviewer’s advice. Incomplete diapause refers to the period when the cocooned larvae develops to the pupal stage, because the cocooned larvae is composed of diapause cocooned larvae and non-diapause cocooned larvae at this time, so it is defined as incomplete diapause.

>>Reviewer’s response: Addressed satisfactorily.

Please provide the details of the method used for diapause incidence estimations.

How do cocooned larval stage durations in Figure 1 translate into the diapause incidence estimates in Figure 2? Were the criteria showed in the Table 1 used to estimate diapause incidence? Cocooned larval stage duration did not exceed 15 days (Figure 1) in any light schedule at any temperature, does it mean the diapause reported in Figure 2 is all incomplete diapause (cocooned larval duration: 6-15 days)?

Response: Thank you for the reviewer’s advice. Diapause incidence = Number of insects that have not entered the pupal stage for more than 15 days/The number of insects tested. The diapause state are used to calculate diapause incidence in Table 1. The diapause was the incidence of complete diapause in Figure 2, with cocooned larvae for more than 15 days.

>>Reviewer’s response: Data in Figure 1 does not look consistent with the one in Figure 2.  If the Cocooned larval stage duration did not exceed 15 days (Figure 1), how can the diapause incidence in Figure 2 be interpreted as complete diapause? As per the criteria mentioned in Table 1 (Cocooned larval stage duration 6-15 days = incomplete diapause; >15 days= complete diapause) diapause incidence in Figure 2 should be incomplete diapause. Please explain.

Average cocooned larval stage duration under 16L:8D at 15oC (Figure 1) is about 12 days. It is not clear how does 12 days average cocooned larval stage duration translate into diapause rate of 0% (Figure 2)?

Response: Thank you for the reviewer’s advice. The diapause rate of cocooned larvae was 0% under 16L:8D at 15, because the cocooned larvae entered the pupal stage in an average time of 12 days, and did not maintain the status of cocooned larvae.

>>Reviewer’s response: At 15oC, the average cocooned larval stage duration is 11-12 days under all the photoperiods (Figure 1). If, the diapause rate of cocooned larvae was 0% under 16L:8D at 15, because all the cocooned larvae entered the pupal stage in an average time of 12 days, how is diapause rate under 10L:14D at 15oC be about 85% (Figure 2)? Interpretation appears incorrect. Please explain.

Transcriptome analysis used the larvae that were in diapause for 15, 25 and 35 days (D15, D25 and D35) under 10L:14D. If there were individuals that remained in cocooned larval stage for 15, 25 and 35 days, why was the average duration of cocooned larval stage only 12-13 days under 10L:14D (Figure 1)? Was the percentage of larvae showing cocooned larval stage duration longer than 15 days very low? Please provide the percentage of larvae that showed cocooned stage longer than 15 days.

Response: Thank you for the reviewer’s advice. The average duration of cocooned larvae stage only 12-13 days is a synthesis of all the data. Some of the insects used in the test entered the pupal stage at 15 days, and most of the non-diapause state entered the pupal stage at 12-13 days, so 12-13 days are shown in Figure 1. When A. aphidimyza enter diapause, they maintain the form of cocooned larvae beyond 15 days, and the proportion of cocooned larvae under 10L:14D photoperiod condition is very high, as can be seen in the diapause rate under 10:14 photoperiod condition in Figure 2 and the diapause rate was as high as 88.7%.

>>Reviewer’s response: This response is not clear. What do you mean by “The average duration of cocooned larvae stage only 12-13 days is a synthesis of all the data.”?

“most of the non-diapause state entered the pupal stage at 12-13 days, so 12-13 days are shown in Figure 1. When A. aphidimyza enter diapause, they maintain the form of cocooned larvae beyond 15 days, and the proportion of cocooned larvae under 10L:14D photoperiod condition is very high, as can be seen in the diapause rate under 10:14 photoperiod condition in Figure 2 and the diapause rate was as high as 88.7%.” So 11.3% individuals did not diapause. Do you mean that average cocooned larval stage duration shown in Figure 1 only include the data from those 11.3 % non-diapausing individuals, and cocooned larval stage durations of diapausing individuals were excluded from the calculation? If yes, please clearly state it in the materials and methods section.

Description of results about differential gene expression is confusing. While comparing genes up- or down- regulated between two groups, say A vs B, please clearly state if the genes are up- or down- regulated (1) in group A compared to B, or (2) in group B compared to A? for example in Figure 4 – ND vs D15, were the 403 genes upregulated in ND or in D15?

Response: Thank you for the reviewer’s advice. We have modified the comparison of gene expression in the article. “The results showed that 1,279 genes showed differential expressed in D15 compared with ND, of which 403 genes were up-regulated and 876 genes were down-regulated. In D25 compared with ND, 1,836 genes showed differential expression, of which 647 were up-regulated and 1,189 were down-regulated. In D35 compared with ND, 1,944 genes showed differential expression, of which 805 were up-regulated and 1,139 were down-regulated. In D25 compared with D15, 190 genes showed differential expression, of which 43 genes were up-regulated and 147 genes were down-regulated. In D35 compared with D15, 835 genes showed differential expression, of which 380 genes were up-regulated and 455 genes were down-regulated. In D35 compared with D25, 246 genes showed differential expression, of which 144 genes were up-regulated and 102 genes were down-regulated. The down-regulated differentially expressed genes were higher than the up-regulated differentially expressed genes in all other treatment groups, except for D25 vs D35.

A total of 2,370 differentially expressed genes were compared among the ND vs D15, ND vs D25, and ND vs D35 groups (Figure 4B). Among them, 778 differentially expressed genes were co-expressed between groups, of these 228 genes were upregulated, and 550 genes were downregulated. There are 213 differentially expressed genes specifically expressed in D15 compared with ND, of which 85 genes were up-regulated and 128 genes were down-regulated. In D25 compared with ND comparison group, there were 356 differentially expressed genes, of which 163 genes were up-regulated and 193 genes were down-regulated; In D35 compared with ND, there were 610 differentially expressed genes, of which 353 genes were up-regulated and 257 genes were down-regulated.

A total of 992 differentially expressed genes were compared among the D15 vs D25, D15 vs D35, and D25 vs D35 groups (Figure 4C). Among them, 4 differentially expressed genes were co-expressed between groups, with 2 genes up-regulated and 2 genes down-regulated. There were 62 differentially expressed genes in D25 compared with D15, of which 12 genes are up-regulated and 50 genes are down-regulated. There were 574 differentially expressed genes in D35 compared with D15, of which 270 genes were up-regulated and 304 genes were down-regulated; 81 differentially expressed genes were specifically expressed in D35 compared with D25, with 49 genes up-regulated and 32 genes down-regulated.” See L284-343

>>Reviewer’s response: Addressed satisfactorily.

Please provide more details about the methods used for estimation of metabolites. Which tissue samples were used for the estimation of metabolite levels – whole body or hemolymph samples? Were the metabolite levels normalized by total protein or weight of the sample to make the estimates comparable across treatments (ND vs D15 vs D25 vs D35)? Y-axis labels for Figure 8 and 9 are ‘relative expression’. For metabolites- ‘level’ is more appropriate word than ‘expression’, please replace ‘expression’ by ‘level’. Please consider using the format e.g. relative level of lipids (%)? Please mention the units of metabolite levels for Figure 8 , 9 in the captions.

Response: Thank you for the reviewer’s advice. The methods used for estimation of metabolites was used whole body and normalized according to total protein, we have modified ‘expression’ to ‘level’ in the article.

>>Reviewer’s response: Addressed satisfactorily.

Line 420-423: “Our results are different from other studies 420 on diapause in aphidimyza [22,23], which may be related to differences in geographical 421 locations. The critical length of diapause in A. aphidimyza was reported to vary at different 422 latitudes and decreases with decreasing latitude [8].” Please provide details about the A. aphidimyza strain such as source location, latitude, year of collection, rearing conditions etc. and if it explains the critical photoperiod observed in the study.

Response: Thank you for the reviewer’s advice. We have supplemented in the material and methods “The tested insects, A. aphidimyza, A. craccivora, were reared in glass greenhouse in Institute of Plant Protection, Shandong Academy of Agricultural SciencesShandong, China, N 36°41’, E 116°54’. Broad bean seedlings were used as host plants to breed Aphis craccivora, which was then used to breed A. aphidimyza. All insect-rearing procedures were performed at 25 , relative humidity of 65%~75%, and photoperiod of 14 hour light and 10 hour dark (14L: 10D). See L114-124

>>Reviewer’s response: Please provide details about the location where the A. aphidimyza strain used in the study was originally collected from wild, latitude of that location, year of collection. Also provide the details of the rearing conditions since collection from wild. Is it an inbred or outbred population?

GO analysis description appears superficial and cluttered. GO terms in Figure 5 are not legible. Separate GO enrichment analysis for up- and down- regulated genes may be more meaningful.

>>Reviewer’s response: Not addressed.

Author Response

Response

This study characterized effects of temperature and photoperiod on the diapause expression, followed by studying the molecular correlates of diapause state. The study does not go further and test the causal role of the candidate genes or metabolites in the development of diapause state. Therefore, “Molecular correlates of diapause in Aphidoletes aphidimyza” may be a more appropriate title of this manuscript.

Response: we have modified the title as “Molecular correlates of diapause in Aphidoletes aphidimyza”. See L1.

>>Reviewer’s response: The word “correlates” is missing from the main title.

Response: Thank you for the reviewer’s advice. we have modified the title as “Molecular correlates of diapause in Aphidoletes aphidimyza”. See L1.

Diapause phenotyping methods have not been described in adequate details in this manuscript. More details may help explain many apparent discrepancies in the diapause data. Please see the specific comments for discrepancies or lack of clarity in the diapause data.

Response: we have see the specific comments for discrepancies or lack of clarity in the diapause data. And have modified the methods.

GO analysis is a very important part of the study. Description of the GO analysis appears very superficial. There is a huge scope to improve the presentation and description of the GO analysis.

Response: we have added some Go analysis. “GO functional annotation was performed on the differentially expressed genes ob-tained by pairwise comparison between the non-diapause group and three different dia-pause groups at different times (Figure 5). The results showed that ND vs D15d, ND vs D25d, ND vs D35d, D15d vs D25d, D15d vs D35d, D25d vs D35d, and D25d vs D35d each annotated 809; 1,129; 1,198; 113; 545 and 151 differentially expressed genes enriched into the three major categories of GO functions, mainly including biological processes, cell composition, and molecular functions. Pairwise comparison at each stage showed that the differentially expressed genes were mainly enriched in the Cell, Membrane, and Organelle of the cell composition; Cellular process, Metabolic pro-cess, and Single-organism process were more abundant in biological processes; Binding, Catalytic activity, and Transporter activity were more abundant in molecular functions.” See L351-355.

Specific comments:

Table 1 shows criteria (developmental stage duration thresholds) used in the study to identify diapause state. What was the basis for using these criteria? Please cite the article if those criteria are based on any previously published study.

Response: Thank you for the reviewer’s advice. We have added the reference about diapause state “[28]Zhai Y.F, Zhang X.Y, Yin Z.J, Zhu Q.S, Tao M, Yu Y, Zheng L. Nymphal Diapause in Laodelphax striatellus (Hemiptera: Delphacidae). Journal of Entomological Science. 2018,53(2):107-122. http://doi.org/10.18474/JES17-140.1 ” See L135

>>Reviewer’s response: Please cite a study in A. aphidimyza where such criteria have been used.

 Response: Until now, no one has studied diapause of A. aphidimyza. We refered to th diapause state “[28]”, and did a lot of preliminary experiments on the division of diapause state in the early stage. We have found that the time for the cocooned larvae to enter the pupal stage did not exceed 15 days no matter what photoperiod conditions, and the cocooned larvae remained in the stage beyond 15 days. Moreover, the development duration of the cocooned larvae was 3-5 days at normal conditions. Therefore this period is used as the critical point for dividing diapause state.

Distinguishing diapause from non-diapause state in aphidimyzais based on the duration of cocooned larval stage. This study used a fixed threshold for cocooned larvae duration (6-15 days: incomplete diapause; >15 days: diapause) to determine if the insect is in diapause or not. The rate of development is highly sensitive to temperature. Typically, low ambient temperatures slow down the rate of development thereby lengthening the duration of all the developmental stages, but slow rate of development need not necessarily induce diapause (developmental arrest). I wonder if a fixed developmental stage duration threshold can be used for a range of temperatures used in this study?

Response: Thank you for the reviewer’s advice. We did a lot of preliminary experiments on the division of diapause state in the early stage. We have found that the time for the cocooned larvae to enter the pupal stage did not exceed 15 days no matter what photoperiod conditions, and the cocooned larvae remained in the stage beyond 15 days. Moreover, the development duration of the cocooned larvae was 3-5 days at normal conditions. Therefore this period is used as the critical point for dividing diapause state.

>>Reviewer’s response: Addressed satisfactorily.

Response: Thank you for your recognition.

3.Please clarify the meaning of incomplete diapause. Is it a real diapause or a non-diapause state with just a slower rate of development? Was the term ‘incomplete diapause’ used for A. aphidimyza in any previously published study? If yes, please cite the article.

Response: Thank you for the reviewer’s advice. Incomplete diapause refers to the period when the cocooned larvae develops to the pupal stage, because the cocooned larvae is composed of diapause cocooned larvae and non-diapause cocooned larvae at this time, so it is defined as incomplete diapause.

>>Reviewer’s response: Addressed satisfactorily.

Response: Thank you for your recognition.

Please provide the details of the method used for diapause incidence estimations.

How do cocooned larval stage durations in Figure 1 translate into the diapause incidence estimates in Figure 2? Were the criteria showed in the Table 1 used to estimate diapause incidence? Cocooned larval stage duration did not exceed 15 days (Figure 1) in any light schedule at any temperature, does it mean the diapause reported in Figure 2 is all incomplete diapause (cocooned larval duration: 6-15 days)?

Response: Thank you for the reviewer’s advice. Diapause incidence = Number of insects that have not entered the pupal stage for more than 15 days/The number of insects tested. The diapause state are used to calculate diapause incidence in Table 1. The diapause was the incidence of complete diapause in Figure 2, with cocooned larvae for more than 15 days.

>>Reviewer’s response: Data in Figure 1 does not look consistent with the one in Figure 2.  If the Cocooned larval stage duration did not exceed 15 days (Figure 1), how can the diapause incidence in Figure 2 be interpreted as complete diapause? As per the criteria mentioned in Table 1 (Cocooned larval stage duration 6-15 days = incomplete diapause; >15 days= complete diapause) diapause incidence in Figure 2 should be incomplete diapause. Please explain.

Response: The diapause rates we investigated were all based on the cocooned larvae for more than 15 days, and those that did not exceed 15 days were considered as non-diapause

Average cocooned larval stage duration under 16L:8D at 15oC (Figure 1) is about 12 days. It is not clear how does 12 days average cocooned larval stage duration translate into diapause rate of 0% (Figure 2)?

Response: Thank you for the reviewer’s advice. The diapause rate of cocooned larvae was 0% under 16L:8D at 15℃, because the cocooned larvae entered the pupal stage in an average time of 12 days, and did not maintain the status of cocooned larvae.

>>Reviewer’s response: At 15oC, the average cocooned larval stage duration is 11-12 days under all the photoperiods (Figure 1). If, the diapause rate of cocooned larvae was 0% under 16L:8D at 15℃, because all the cocooned larvae entered the pupal stage in an average time of 12 days, how is diapause rate under 10L:14D at 15oC be about 85% (Figure 2)? Interpretation appears incorrect. Please explain.

Response: in Figure 1, We only counted the developmental duration of the non-diapause period. If the diapause period is entered, the diapause time can reach more than 60 days, which is not statistically significant.

 Response: in Figure1, the develoumental duration

Transcriptome analysis used the larvae that were in diapause for 15, 25 and 35 days (D15, D25 and D35) under 10L:14D. If there were individuals that remained in cocooned larval stage for 15, 25 and 35 days, why was the average duration of cocooned larval stage only 12-13 days under 10L:14D (Figure 1)? Was the percentage of larvae showing cocooned larval stage duration longer than 15 days very low? Please provide the percentage of larvae that showed cocooned stage longer than 15 days.

Response: Thank you for the reviewer’s advice. The average duration of cocooned larvae stage only 12-13 days is a synthesis of all the data. Some of the insects used in the test entered the pupal stage at 15 days, and most of the non-diapause state entered the pupal stage at 12-13 days, so 12-13 days are shown in Figure 1. When A. aphidimyza enter diapause, they maintain the form of cocooned larvae beyond 15 days, and the proportion of cocooned larvae under 10L:14D photoperiod condition is very high, as can be seen in the diapause rate under 10:14 photoperiod condition in Figure 2 and the diapause rate was as high as 88.7%.

>>Reviewer’s response: This response is not clear. What do you mean by “The average duration of cocooned larvae stage only 12-13 days is a synthesis of all the data.”?

“most of the non-diapause state entered the pupal stage at 12-13 days, so 12-13 days are shown in Figure 1. When A. aphidimyza enter diapause, they maintain the form of cocooned larvae beyond 15 days, and the proportion of cocooned larvae under 10L:14D photoperiod condition is very high, as can be seen in the diapause rate under 10:14 photoperiod condition in Figure 2 and the diapause rate was as high as 88.7%.” So 11.3% individuals did not diapause. Do you mean that average cocooned larval stage duration shown in Figure 1 only include the data from those 11.3 % non-diapausing individuals, and cocooned larval stage durations of diapausing individuals were excluded from the calculation? If yes, please clearly state it in the materials and methods section.

Response: in Figure 1, We only counted the developmental duration of the non-diapause period. If the diapause period is entered, the diapause time can reach more than 60 days, which is not statistically significant.

Description of results about differential gene expression is confusing. While comparing genes up- or down- regulated between two groups, say A vs B, please clearly state if the genes are up- or down- regulated (1) in group A compared to B, or (2) in group B compared to A? for example in Figure 4 – ND vs D15, were the 403 genes upregulated in ND or in D15?

Response: Thank you for the reviewer’s advice. We have modified the comparison of gene expression in the article. “The results showed that 1,279 genes showed differential expressed in D15 compared with ND, of which 403 genes were up-regulated and 876 genes were down-regulated. In D25 compared with ND, 1,836 genes showed differential expression, of which 647 were up-regulated and 1,189 were down-regulated. In D35 compared with ND, 1,944 genes showed differential expression, of which 805 were up-regulated and 1,139 were down-regulated. In D25 compared with D15, 190 genes showed differential expression, of which 43 genes were up-regulated and 147 genes were down-regulated. In D35 compared with D15, 835 genes showed differential expression, of which 380 genes were up-regulated and 455 genes were down-regulated. In D35 compared with D25, 246 genes showed differential expression, of which 144 genes were up-regulated and 102 genes were down-regulated. The down-regulated differentially expressed genes were higher than the up-regulated differentially expressed genes in all other treatment groups, except for D25 vs D35.

A total of 2,370 differentially expressed genes were compared among the ND vs D15, ND vs D25, and ND vs D35 groups (Figure 4B). Among them, 778 differentially expressed genes were co-expressed between groups, of these 228 genes were upregulated, and 550 genes were downregulated. There are 213 differentially expressed genes specifically expressed in D15 compared with ND, of which 85 genes were up-regulated and 128 genes were down-regulated. In D25 compared with ND comparison group, there were 356 differentially expressed genes, of which 163 genes were up-regulated and 193 genes were down-regulated; In D35 compared with ND, there were 610 differentially expressed genes, of which 353 genes were up-regulated and 257 genes were down-regulated.

A total of 992 differentially expressed genes were compared among the D15 vs D25, D15 vs D35, and D25 vs D35 groups (Figure 4C). Among them, 4 differentially expressed genes were co-expressed between groups, with 2 genes up-regulated and 2 genes down-regulated. There were 62 differentially expressed genes in D25 compared with D15, of which 12 genes are up-regulated and 50 genes are down-regulated. There were 574 differentially expressed genes in D35 compared with D15, of which 270 genes were up-regulated and 304 genes were down-regulated; 81 differentially expressed genes were specifically expressed in D35 compared with D25, with 49 genes up-regulated and 32 genes down-regulated.” See L284-343

>>Reviewer’s response: Addressed satisfactorily.

 Response: Thank you for your recognition.

Please provide more details about the methods used for estimation of metabolites. Which tissue samples were used for the estimation of metabolite levels – whole body or hemolymph samples? Were the metabolite levels normalized by total protein or weight of the sample to make the estimates comparable across treatments (ND vs D15 vs D25 vs D35)? Y-axis labels for Figure 8 and 9 are ‘relative expression’. For metabolites- ‘level’ is more appropriate word than ‘expression’, please replace ‘expression’ by ‘level’. Please consider using the format e.g. relative level of lipids (%)? Please mention the units of metabolite levels for Figure 8 , 9 in the captions.

Response: Thank you for the reviewer’s advice. The methods used for estimation of metabolites was used whole body and normalized according to total protein, we have modified ‘expression’ to ‘level’ in the article.

>>Reviewer’s response: Addressed satisfactorily.

Response: Thank you for your recognition.

Line 420-423: “Our results are different from other studies 420 on diapause in aphidimyza [22,23], which may be related to differences in geographical 421 locations. The critical length of diapause in A. aphidimyza was reported to vary at different 422 latitudes and decreases with decreasing latitude [8].” Please provide details about the A. aphidimyza strain such as source location, latitude, year of collection, rearing conditions etc. and if it explains the critical photoperiod observed in the study.

Response: Thank you for the reviewer’s advice. We have supplemented in the material and methods “The tested insects, A. aphidimyza, A. craccivora, were reared in glass greenhouse in Institute of Plant Protection, Shandong Academy of Agricultural Sciences(Shandong, China, N 36°41’, E 116°54’). Broad bean seedlings were used as host plants to breed Aphis craccivora, which was then used to breed A. aphidimyza. All insect-rearing procedures were performed at 25 ℃, relative humidity of 65%~75%, and photoperiod of 14 hour light and 10 hour dark (14L: 10D).” See L114-124

>>Reviewer’s response: Please provide details about the location where the A. aphidimyza strain used in the study was originally collected from wild, latitude of that location, year of collection. Also provide the details of the rearing conditions since collection from wild. Is it an inbred or outbred population?

Response: We have added it as“The tested insects, A. aphidimyza and A. Craccivora were collected from pea crop fields in Jinan, Shandong Province, China (N 36°41’, E 116°54’), and were reared in insect rearing cages of glass greenhouse (26±1℃, 70±5% RH, 14L:10D) in Institute of Plant Protection, Shandong Academy of Agricultural Sciences(Shandong,  China, N 36°41’, E 116°54’).”

GO analysis description appears superficial and cluttered. GO terms in Figure 5 are not legible. Separate GO enrichment analysis for up- and down- regulated genes may be more meaningful.

>>Reviewer’s response: Not addressed.

Response: In Figures SIII to SVIII, we have separated GO enrichment analysis for up- and down- regulated genes. And added some analysis “The significantly enriched entries (P<0.05) were mainly related to Metabolic and Cel-lular process in the six groups of comparisons, wit the first 20 GO entries detailed in ribo-some biogenesis、integral component of membrane、sequence-specific DNA binding (Figures SIII to SVIII).” See L360-364.

Round 3

Reviewer 3 Report

Comments and Suggestions for Authors

Reviewer response to 2nd revision; responses are highlighted in RED

This study characterized effects of temperature and photoperiod on the diapause expression, followed by studying the molecular correlates of diapause state. The study does not go further and test the causal role of the candidate genes or metabolites in the development of diapause state. Therefore, “Molecular correlates of diapause in Aphidoletes aphidimyza” may be a more appropriate title of this manuscript.

Response: we have modified the title as “Molecular correlates of diapause in Aphidoletes aphidimyza”. See L1.

>>Reviewer’s response: The word “correlates” is missing from the main title.

Response: Thank you for the reviewer’s advice. we have modified the title as “Molecular correlates of diapause in Aphidoletes aphidimyza”. See L1.

>>Reviewer’s response: Addressed

Diapause phenotyping methods have not been described in adequate details in this manuscript. More details may help explain many apparent discrepancies in the diapause data. Please see the specific comments for discrepancies or lack of clarity in the diapause data.

Response: we have see the specific comments for discrepancies or lack of clarity in the diapause data. And have modified the methods.

GO analysis is a very important part of the study. Description of the GO analysis appears very superficial. There is a huge scope to improve the presentation and description of the GO analysis.

Response: we have added some Go analysis. “GO functional annotation was performed on the differentially expressed genes ob-tained by pairwise comparison between the non-diapause group and three different dia-pause groups at different times (Figure 5). The results showed that ND vs D15d, ND vs D25d, ND vs D35d, D15d vs D25d, D15d vs D35d, D25d vs D35d, and D25d vs D35d each annotated 809; 1,129; 1,198; 113; 545 and 151 differentially expressed genes enriched into the three major categories of GO functions, mainly including biological processes, cell composition, and molecular functions. Pairwise comparison at each stage showed that the differentially expressed genes were mainly enriched in the Cell, Membrane, and Organelle of the cell composition; Cellular process, Metabolic pro-cess, and Single-organism process were more abundant in biological processes; Binding, Catalytic activity, and Transporter activity were more abundant in molecular functions.” See L351-355.

Specific comments:

Table 1 shows criteria (developmental stage duration thresholds) used in the study to identify diapause state. What was the basis for using these criteria? Please cite the article if those criteria are based on any previously published study.

Response: Thank you for the reviewer’s advice. We have added the reference about diapause state “[28]Zhai Y.F, Zhang X.Y, Yin Z.J, Zhu Q.S, Tao M, Yu Y, Zheng L. Nymphal Diapause in Laodelphax striatellus (Hemiptera: Delphacidae). Journal of Entomological Science. 2018,53(2):107-122. http://doi.org/10.18474/JES17-140.1 ” See L135

>>Reviewer’s response: Please cite a study in A. aphidimyza where such criteria have been used.

 Response: Until now, no one has studied diapause of A. aphidimyza. We refered to th diapause state “[28]”, and did a lot of preliminary experiments on the division of diapause state in the early stage. We have found that the time for the cocooned larvae to enter the pupal stage did not exceed 15 days no matter what photoperiod conditions, and the cocooned larvae remained in the stage beyond 15 days. Moreover, the development duration of the cocooned larvae was 3-5 days at normal conditions. Therefore this period is used as the critical point for dividing diapause state.

>>Reviewer’s response: There are already many papers on A. aphidimyza diapause, so it is not correct to say that no one has studied diapause of A. aphidimyza. I believe, the type of diapause state categorization done in this study is being reported for the first time in A. aphidimyza. So, it will be important to include the data from your preliminary experiments in supplementary figues. Please plot the development time on X-axis and frequencies for each development time category on Y-axis. Additionally, please provide stacked bar graph showing percentage of individuals showing non-diapause, incomplete diapause and diapause state for each light and temperature condition in Figure 2.

Distinguishing diapause from non-diapause state in aphidimyzais based on the duration of cocooned larval stage. This study used a fixed threshold for cocooned larvae duration (6-15 days: incomplete diapause; >15 days: diapause) to determine if the insect is in diapause or not. The rate of development is highly sensitive to temperature. Typically, low ambient temperatures slow down the rate of development thereby lengthening the duration of all the developmental stages, but slow rate of development need not necessarily induce diapause (developmental arrest). I wonder if a fixed developmental stage duration threshold can be used for a range of temperatures used in this study?

Response: Thank you for the reviewer’s advice. We did a lot of preliminary experiments on the division of diapause state in the early stage. We have found that the time for the cocooned larvae to enter the pupal stage did not exceed 15 days no matter what photoperiod conditions, and the cocooned larvae remained in the stage beyond 15 days. Moreover, the development duration of the cocooned larvae was 3-5 days at normal conditions. Therefore this period is used as the critical point for dividing diapause state.

>>Reviewer’s response: Addressed satisfactorily.

Response: Thank you for your recognition.

3.Please clarify the meaning of incomplete diapause. Is it a real diapause or a non-diapause state with just a slower rate of development? Was the term ‘incomplete diapause’ used for A. aphidimyza in any previously published study? If yes, please cite the article.

 Response: Thank you for the reviewer’s advice. Incomplete diapause refers to the period when the cocooned larvae develops to the pupal stage, because the cocooned larvae is composed of diapause cocooned larvae and non-diapause cocooned larvae at this time, so it is defined as incomplete diapause.

>>Reviewer’s response: Addressed satisfactorily.

Response: Thank you for your recognition.

Please provide the details of the method used for diapause incidence estimations.

How do cocooned larval stage durations in Figure 1 translate into the diapause incidence estimates in Figure 2? Were the criteria showed in the Table 1 used to estimate diapause incidence? Cocooned larval stage duration did not exceed 15 days (Figure 1) in any light schedule at any temperature, does it mean the diapause reported in Figure 2 is all incomplete diapause (cocooned larval duration: 6-15 days)?

Response: Thank you for the reviewer’s advice. Diapause incidence = Number of insects that have not entered the pupal stage for more than 15 days/The number of insects tested. The diapause state are used to calculate diapause incidence in Table 1. The diapause was the incidence of complete diapause in Figure 2, with cocooned larvae for more than 15 days.

>>Reviewer’s response: Data in Figure 1 does not look consistent with the one in Figure 2.  If the Cocooned larval stage duration did not exceed 15 days (Figure 1), how can the diapause incidence in Figure 2 be interpreted as complete diapause? As per the criteria mentioned in Table 1 (Cocooned larval stage duration 6-15 days = incomplete diapause; >15 days= complete diapause) diapause incidence in Figure 2 should be incomplete diapause. Please explain.

Response: The diapause rates we investigated were all based on the cocooned larvae for more than 15 days, and those that did not exceed 15 days were considered as non-diapause

Average cocooned larval stage duration under 16L:8D at 15oC (Figure 1) is about 12 days. It is not clear how does 12 days average cocooned larval stage duration translate into diapause rate of 0% (Figure 2)?

Response: Thank you for the reviewer’s advice. The diapause rate of cocooned larvae was 0% under 16L:8D at 15, because the cocooned larvae entered the pupal stage in an average time of 12 days, and did not maintain the status of cocooned larvae.

>>Reviewer’s response: At 15oC, the average cocooned larval stage duration is 11-12 days under all the photoperiods (Figure 1). If, the diapause rate of cocooned larvae was 0% under 16L:8D at 15, because all the cocooned larvae entered the pupal stage in an average time of 12 days, how is diapause rate under 10L:14D at 15oC be about 85% (Figure 2)? Interpretation appears incorrect. Please explain.

Response: in Figure 1, We only counted the developmental duration of the non-diapause period. If the diapause period is entered, the diapause time can reach more than 60 days, which is not statistically significant.

>>Reviewer’s response: Please clearly provide the development time calculation method (that only non-diapause larvae were included in development time estimation) in materials and methods section or in the Figure 1 caption

Transcriptome analysis used the larvae that were in diapause for 15, 25 and 35 days (D15, D25 and D35) under 10L:14D. If there were individuals that remained in cocooned larval stage for 15, 25 and 35 days, why was the average duration of cocooned larval stage only 12-13 days under 10L:14D (Figure 1)? Was the percentage of larvae showing cocooned larval stage duration longer than 15 days very low? Please provide the percentage of larvae that showed cocooned stage longer than 15 days.

Response: Thank you for the reviewer’s advice. The average duration of cocooned larvae stage only 12-13 days is a synthesis of all the data. Some of the insects used in the test entered the pupal stage at 15 days, and most of the non-diapause state entered the pupal stage at 12-13 days, so 12-13 days are shown in Figure 1. When A. aphidimyza enter diapause, they maintain the form of cocooned larvae beyond 15 days, and the proportion of cocooned larvae under 10L:14D photoperiod condition is very high, as can be seen in the diapause rate under 10:14 photoperiod condition in Figure 2 and the diapause rate was as high as 88.7%.

>>Reviewer’s response: This response is not clear. What do you mean by “The average duration of cocooned larvae stage only 12-13 days is a synthesis of all the data.”? Please provide stacked bar graph showing percentage of individuals showing non-diapause, incomplete diapause and diapause state for each light and temperature condition in Figure 2.

“most of the non-diapause state entered the pupal stage at 12-13 days, so 12-13 days are shown in Figure 1. When A. aphidimyza enter diapause, they maintain the form of cocooned larvae beyond 15 days, and the proportion of cocooned larvae under 10L:14D photoperiod condition is very high, as can be seen in the diapause rate under 10:14 photoperiod condition in Figure 2 and the diapause rate was as high as 88.7%.” So 11.3% individuals did not diapause. Do you mean that average cocooned larval stage duration shown in Figure 1 only include the data from those 11.3 % non-diapausing individuals, and cocooned larval stage durations of diapausing individuals were excluded from the calculation? If yes, please clearly state it in the materials and methods section.

Response: in Figure 1, We only counted the developmental duration of the non-diapause period. If the diapause period is entered, the diapause time can reach more than 60 days, which is not statistically significant.

>>Reviewer’s response: Please clearly specify the calculation method in materials and methods section or in the Figure 1 caption

 Description of results about differential gene expression is confusing. While comparing genes up- or down- regulated between two groups, say A vs B, please clearly state if the genes are up- or down- regulated (1) in group A compared to B, or (2) in group B compared to A? for example in Figure 4 – ND vs D15, were the 403 genes upregulated in ND or in D15?

Response: Thank you for the reviewer’s advice. We have modified the comparison of gene expression in the article. “The results showed that 1,279 genes showed differential expressed in D15 compared with ND, of which 403 genes were up-regulated and 876 genes were down-regulated. In D25 compared with ND, 1,836 genes showed differential expression, of which 647 were up-regulated and 1,189 were down-regulated. In D35 compared with ND, 1,944 genes showed differential expression, of which 805 were up-regulated and 1,139 were down-regulated. In D25 compared with D15, 190 genes showed differential expression, of which 43 genes were up-regulated and 147 genes were down-regulated. In D35 compared with D15, 835 genes showed differential expression, of which 380 genes were up-regulated and 455 genes were down-regulated. In D35 compared with D25, 246 genes showed differential expression, of which 144 genes were up-regulated and 102 genes were down-regulated. The down-regulated differentially expressed genes were higher than the up-regulated differentially expressed genes in all other treatment groups, except for D25 vs D35.

A total of 2,370 differentially expressed genes were compared among the ND vs D15, ND vs D25, and ND vs D35 groups (Figure 4B). Among them, 778 differentially expressed genes were co-expressed between groups, of these 228 genes were upregulated, and 550 genes were downregulated. There are 213 differentially expressed genes specifically expressed in D15 compared with ND, of which 85 genes were up-regulated and 128 genes were down-regulated. In D25 compared with ND comparison group, there were 356 differentially expressed genes, of which 163 genes were up-regulated and 193 genes were down-regulated; In D35 compared with ND, there were 610 differentially expressed genes, of which 353 genes were up-regulated and 257 genes were down-regulated.

A total of 992 differentially expressed genes were compared among the D15 vs D25, D15 vs D35, and D25 vs D35 groups (Figure 4C). Among them, 4 differentially expressed genes were co-expressed between groups, with 2 genes up-regulated and 2 genes down-regulated. There were 62 differentially expressed genes in D25 compared with D15, of which 12 genes are up-regulated and 50 genes are down-regulated. There were 574 differentially expressed genes in D35 compared with D15, of which 270 genes were up-regulated and 304 genes were down-regulated; 81 differentially expressed genes were specifically expressed in D35 compared with D25, with 49 genes up-regulated and 32 genes down-regulated.” See L284-343

>>Reviewer’s response: Addressed satisfactorily.

 Response: Thank you for your recognition.

Please provide more details about the methods used for estimation of metabolites. Which tissue samples were used for the estimation of metabolite levels – whole body or hemolymph samples? Were the metabolite levels normalized by total protein or weight of the sample to make the estimates comparable across treatments (ND vs D15 vs D25 vs D35)? Y-axis labels for Figure 8 and 9 are ‘relative expression’. For metabolites- ‘level’ is more appropriate word than ‘expression’, please replace ‘expression’ by ‘level’. Please consider using the format e.g. relative level of lipids (%)? Please mention the units of metabolite levels for Figure 8 , 9 in the captions.

Response: Thank you for the reviewer’s advice. The methods used for estimation of metabolites was used whole body and normalized according to total protein, we have modified ‘expression’ to ‘level’ in the article.

>>Reviewer’s response: Addressed satisfactorily.

Response: Thank you for your recognition.

Line 420-423: “Our results are different from other studies 420 on diapause in aphidimyza [22,23], which may be related to differences in geographical 421 locations. The critical length of diapause in A. aphidimyza was reported to vary at different 422 latitudes and decreases with decreasing latitude [8].” Please provide details about the A. aphidimyza strain such as source location, latitude, year of collection, rearing conditions etc. and if it explains the critical photoperiod observed in the study.

Response: Thank you for the reviewer’s advice. We have supplemented in the material and methods “The tested insects, A. aphidimyza, A. craccivora, were reared in glass greenhouse in Institute of Plant Protection, Shandong Academy of Agricultural SciencesShandong, China, N 36°41’, E 116°54’. Broad bean seedlings were used as host plants to breed Aphis craccivora, which was then used to breed A. aphidimyza. All insect-rearing procedures were performed at 25 , relative humidity of 65%~75%, and photoperiod of 14 hour light and 10 hour dark (14L: 10D). See L114-124

>>Reviewer’s response: Please provide details about the location where the A. aphidimyza strain used in the study was originally collected from wild, latitude of that location, year of collection. Also provide the details of the rearing conditions since collection from wild. Is it an inbred or outbred population?

Response: We have added it as“The tested insects, A. aphidimyza and A. Craccivora were collected from pea crop fields in Jinan, Shandong Province, China (N 36°41’, E 116°54’), and were reared in insect rearing cages of glass greenhouse (26±1, 70±5% RH, 14L:10D) in Institute of Plant Protection, Shandong Academy of Agricultural SciencesShandong,  China, N 36°41’, E 116°54’.”

>>Reviewer’s response: Addressed satisfactorily.

GO analysis description appears superficial and cluttered. GO terms in Figure 5 are not legible. Separate GO enrichment analysis for up- and down- regulated genes may be more meaningful.

>>Reviewer’s response: Not addressed.

Response: In Figures SIII to SVIII, we have separated GO enrichment analysis for up- and down- regulated genes. And added some analysis “The significantly enriched entries (P<0.05) were mainly related to Metabolic and Cel-lular process in the six groups of comparisons, wit the first 20 GO entries detailed in ribo-some biogenesisintegral component of membranesequence-specific DNA binding (Figures SIII to SVIII).” See L360-364.

>>Reviewer’s response: Addressed satisfactorily.

Author Response

Response to Reviewer 1, 2 and 3 Comments

Comments 1: Table 1 shows criteria (developmental stage duration thresholds) used in the study to identify diapause state. What was the basis for using these criteria? Please cite the article if those criteria are based on any previously published study.

Response: Thank you for the reviewer’s advice. We have added the reference about diapause state “[28]Zhai Y.F, Zhang X.Y, Yin Z.J, Zhu Q.S, Tao M, Yu Y, Zheng L. Nymphal Diapause in Laodelphax striatellus (Hemiptera: Delphacidae). Journal of Entomological Science. 2018,53(2):107-122. http://doi.org/10.18474/JES17-140.1 ” See L135

>>Reviewer’s response: Please cite a study in A. aphidimyza where such criteria have been used.

 Response: Until now, no one has studied diapause of A. aphidimyza. We refered to th diapause state “[28]”, and did a lot of preliminary experiments on the division of diapause state in the early stage. We have found that the time for the cocooned larvae to enter the pupal stage did not exceed 15 days no matter what photoperiod conditions, and the cocooned larvae remained in the stage beyond 15 days. Moreover, the development duration of the cocooned larvae was 3-5 days at normal conditions. Therefore this period is used as the critical point for dividing diapause state.

>>Reviewer’s response: There are already many papers on A. aphidimyza diapause, so it is not correct to say that no one has studied diapause of A. aphidimyza. I believe, the type of diapause state categorization done in this study is being reported for the first time in A. aphidimyza. So, it will be important to include the data from your preliminary experiments in supplementary figues. Please plot the development time on X-axis and frequencies for each development time category on Y-axis. Additionally, please provide stacked bar graph showing percentage of individuals showing non-diapause, incomplete diapause and diapause state for each light and temperature condition in Figure 2.

Response: Thank you for the reviewer’s advice. I don't think it makes much sense to count the rate of incomplete diapause, because the rate of incomplete diapause is actually the rate of insects entering diapause. Personally, I think the diapause rate in Fig. 2 is a stacked bar graph which is more suitable than the line graph.

Comments 2: Average cocooned larval stage duration under 16L:8D at 15oC (Figure 1) is about 12 days. It is not clear how does 12 days average cocooned larval stage duration translate into diapause rate of 0% (Figure 2)?

Response: Thank you for the reviewer’s advice. The diapause rate of cocooned larvae was 0% under 16L:8D at 15℃, because the cocooned larvae entered the pupal stage in an average time of 12 days, and did not maintain the status of cocooned larvae.

>>Reviewer’s response: At 15oC, the average cocooned larval stage duration is 11-12 days under all the photoperiods (Figure 1). If, the diapause rate of cocooned larvae was 0% under 16L:8D at 15℃, because all the cocooned larvae entered the pupal stage in an average time of 12 days, how is diapause rate under 10L:14D at 15oC be about 85% (Figure 2)? Interpretation appears incorrect. Please explain.

Response: in Figure 1, We only counted the developmental duration of the non-diapause period. If the diapause period is entered, the diapause time can reach more than 60 days, which is not statistically significant.

>>Reviewer’s response: Please clearly provide the development time calculation method (that only non-diapause larvae were included in development time estimation) in materials and methods section or in the Figure 1 caption

Response: Thank you for the reviewer’s advice. We have added the annotation in Figure 1. “The developmental periods of non-diapause insects was selected as the developmental periods of A. aphidimyza under different temperature and photoperiod conditions. And developmental periods = Sum of developmental periods of non-diapause insects/ Number of non-diapause insects.” See L211-214

Comments 3: Transcriptome analysis used the larvae that were in diapause for 15, 25 and 35 days (D15, D25 and D35) under 10L:14D. If there were individuals that remained in cocooned larval stage for 15, 25 and 35 days, why was the average duration of cocooned larval stage only 12-13 days under 10L:14D (Figure 1)? Was the percentage of larvae showing cocooned larval stage duration longer than 15 days very low? Please provide the percentage of larvae that showed cocooned stage longer than 15 days.

Response: Thank you for the reviewer’s advice. The average duration of cocooned larvae stage only 12-13 days is a synthesis of all the data. Some of the insects used in the test entered the pupal stage at 15 days, and most of the non-diapause state entered the pupal stage at 12-13 days, so 12-13 days are shown in Figure 1. When A. aphidimyza enter diapause, they maintain the form of cocooned larvae beyond 15 days, and the proportion of cocooned larvae under 10L:14D photoperiod condition is very high, as can be seen in the diapause rate under 10:14 photoperiod condition in Figure 2 and the diapause rate was as high as 88.7%.

>>Reviewer’s response: This response is not clear. What do you mean by “The average duration of cocooned larvae stage only 12-13 days is a synthesis of all the data.”? Please provide stacked bar graph showing percentage of individuals showing non-diapause, incomplete diapause and diapause state for each light and temperature condition in Figure 2.

Response: Thank you for the reviewer’s advice. The developmental period in Figure 1 is a comprehensive calculation of the developmental time of all insects that have not entered diapause. The actual 12-13 days in the incomplete diapause period is an average calculation of the developmental time of all insects that have not entered diapause at this stage. Because such data were chosen, it was possible to visualize the effects of different temperatures and photoperiods on developmental time.

Comments 4: most of the non-diapause state entered the pupal stage at 12-13 days, so 12-13 days are shown in Figure 1. When A. aphidimyza enter diapause, they maintain the form of cocooned larvae beyond 15 days, and the proportion of cocooned larvae under 10L:14D photoperiod condition is very high, as can be seen in the diapause rate under 10:14 photoperiod condition in Figure 2 and the diapause rate was as high as 88.7%.” So 11.3% individuals did not diapause. Do you mean that average cocooned larval stage duration shown in Figure 1 only include the data from those 11.3 % non-diapausing individuals, and cocooned larval stage durations of diapausing individuals were excluded from the calculation? If yes, please clearly state it in the materials and methods section.

Response: in Figure 1, We only counted the developmental duration of the non-diapause period. If the diapause period is entered, the diapause time can reach more than 60 days, which is not statistically significant.

>>Reviewer’s response: Please clearly specify the calculation method in materials and methods section or in the Figure 1 caption

Response: Thank you for the reviewer’s advice. We have added the annotation in Figure 1. “The developmental periods of non-diapause insects was selected as the developmental periods of A. aphidimyza under different temperature and photoperiod conditions. And developmental periods = Sum of developmental periods of non-diapause insects/ Number of non-diapause insects.” See L211-214

Round 4

Reviewer 3 Report

Comments and Suggestions for Authors

Lines 211-214 are not making sense. Please simply write -

Only non-diapausing larvae were included in the estimation of development duration.

Author Response

Response to Reviewer 3 Comments

Lines 211-214 are not making sense. Please simply write -

Only non-diapausing larvae were included in the estimation of development duration.

Response: Thank you for the reviewer’s valuable advice. We have modified it as “non-diapausing larvae were included in the estimation of development duration.”